



# Partitioning of carbon export in the upper water column of the oligotrophic South China Sea

Yifan Ma[1], Kuanbo Zhou[1]. Weifang Chen[1], Junhui Chen[1], Jin-Yu Terence Yang[1] & Minhan Dai[1]

[1]State Key Laboratory of Marine Environmental Science, College of Ocean and Earth Sciences, Xiamen University, Xiamen, 361102, China

*Correspondence to*: Minhan Dai (mdai@xmu.edu.cn)

**Abstract.** We conducted high vertical resolution samplings of total and particulate $^{234}$Th along with particulate organic carbon (POC) in the summer of 2017 to examine nutrient-dependent structures of export productivity within the euphotic zone (Ez) of the oligotrophic basin of the South China Sea (SCS). Nitrate concentrations throughout the study area were below detection limit in the nutrient-depleted layer (NDL) above the nutricline, while they sharply increased with depth in the nutrient-replete layer (NRL) across the nutricline until the base of the Ez. Based on our high resolution vertical profilings of $^{234}$Th/$^{238}$U disequilibria, this study for the first time estimated POC export fluxes both out of the NDL and at the horizon of the Ez base. Total $^{234}$Th deficit relative to $^{238}$U occurred in the NDL at all study sites, while $^{234}$Th was mostly in equilibrium with $^{238}$U in the NRL except at the northmost station SEATS (116º E, 18º N), where the $^{234}$Th deficit could also be observed in the NRL. By combining 1D steady-state $^{234}$Th fluxes and POC/$^{234}$Th ratios, we derived vertical patterns of POC export fluxes. Values were 1.6±0.6 mmol C m$^{-2}$ d$^{-1}$ at the NDL base, representing approximately half of the flux estimated at the base of the Ez at station SEATS; for the rest of the sampling sites, POC export fluxes at the NDL base (averaged at 2.3±1.1 mmol C m$^{-2}$ d$^{-1}$) were identical within error to those at the base of the Ez (1.9±0.5 mmol C m$^{-2}$ d$^{-1}$), suggesting rapid export of POC out of the NDL. This finding fundamentally changes our traditional view that the NDL, being depleted in nutrients, would not be a net exporter of POC. Based on the positive relationship between POC export fluxes at the NDL base and supply potential of subsurface nutrients (i.e., nutricline depth and nutrient concentrations), we found that POC export fluxes (averaged at 3.4±1.2 mmol C m$^{-2}$ d$^{-1}$) at the NDL base at stations with shallow nutriclines and high subsurface nutrient concentrations were ~100% higher than the fluxes (averaged at 1.6±0.5 mmol C m$^{-2}$ d$^{-1}$) at other stations. We used a two-endmember mixing model based on the mass and $^{15}$N-isotopic balances to further evaluate the potential sources of new nitrogen that could support the observed particle export at stations SEATS and SS1, located respectively in the northern and southern basin of the SCS with different hydrological features. We showed that more than 50% of the particle flux out of the NDL was supported by nitrate sources other than atmospheric deposition and nitrogen fixation: likely supply from depth associated with episodic intrusions. However, the exact mechanisms and pathways for subsurface nutrients to support the export production from the NDL merit careful and dedicated studies.

**Keywords**: Export productivity, nutrient-depleted layer, $^{234}$Th/$^{238}$U disequilibrium, the South China Sea



## 1 Introduction

The marine biological carbon pump (BCP) plays a central role in sequestrating atmospheric $CO_2$, thereby mitigating human-
induced climate change. Despite great efforts that have been devoted to studying the BCP, there remains critical knowledge
gaps in its structure, function and efficiency (Siegel et al., 2020). Recently, the EXPORTS (EXport Processes in the Ocean
from RemoTe Sensing) program has implemented comprehensive experiments which examine export flux pathways, plankton
community composition, food web processes, and biogeochemical properties of the ecosystem, in order to achieve an improved
understanding of export fluxes and the BCP (Siegel et al., 2016; 2020).

Among other factors, depth-dependent particle export at different horizons within the euphotic zone (Ez), and how these
exports are sustained by different nutrients sources, remains largely unknown. Most previous studies have treated the Ez as a
single box and chose a fixed depth (e.g., 100 or 150 m) as the export horizon. A recent study has suggested that using a fixed
depth instead of the *in situ* Ez depth as the export horizon would lead to the magnitude of global POC export flux to be
underestimated by a factor of two (Buesseler et al., 2020a). In the oligotrophic oceans, permanent stratification limits nutrient
supply from depth; the Ez thus could be divided into a two-layer structure based on nutrient concentrations: (1) the Nutrient-
depleted Layer (NDL) between the ocean surface and the top of the nutricline, and (2) the Nutrient-replete Layer between the
nutricline and the base of the Ez (Du et al., 2017). Conventional concepts suggest that regenerated nutrients predominantly
support biological productivity in the NDL where export production is limited due to the absence of new nutrient supplies
(Eppley and Peterson, 1979; Goldman, 1984). Meanwhile, Coale and Bruland (1987) noticed layerred structure of $^{234}$Th-$^{238}$U
disequilibria in the Ez, composed of an upper oligotrophic layer characterized by low new production values, low net
scavenging; and a subsurface eutrophic layer with higher new production values, and suggested that new production rather
than total primary production dertemined the scavenging of the reactive elements such as $^{234}$Th.

Recently, some observations have however indicated that particles sourced from surface waters with extremely low nutrient
concentrations may substantially contribute to the downward fluxes at depth. Scharek et al. (1999) observed that the diatom-
diazotroph assemblages (*H.hauckii* contained *Richelia*-type endosymbionts with heterocysts) in the surface nutrient-deficient
mixed layer dominated downward particle fluxes collected by a sediment trap at 150 m depth at the oligotrophic station
ALOHA (158° W, 22° N). Liu et al. (2007) observed consistent $\delta^{13}C_{POC}$ values between sediment trap samples collected at 100
m and suspended particles in the upper 20 m in the South China Sea (SCS) basin, suggesting that particles in the trap
predominantly originated from the upper ocean (i.e., 20 m). The ecosystems of nutrient-depleted surface waters may play an
important role in carbon export. Different pathways to introduce new nutrients have been suggested to support the carbon
export from the NDL; for example, high rates of nitrogen fixation (NF) in the NDL could support 26-47% of the particle fluxes
at station ALOHA (Böttjer et al., 2017). In addition, episodic eddy events that uplift the nutricline and deliver deep stocks of
nutrients to the NDL might also contribute to POC export fluxes from the upper ocean (Johnson et al., 2010). Nevertheless, it
remains unclear how the nutricline shift controls nutrient supply to the surface waters and affects the downward POC export
flux at the NDL and Ez horizons.





The semi-enclosed South China Sea (SCS), the largest marginal sea in the western North Pacific Ocean, is characterized as an oligotrophic basin due to intensive stratification (Du et al., 2017). High phytoplankton diversity and primary production rates (Xie et al., 2018) are, nevertheless, still observed in this nutrient-deficit ecosystem.

Several previous studies quantified the [234]Th-based POC export flux and explored the mechanisms controlling export in the SCS. Seasonally, POC export fluxes are elevated in winter driven by the deepening of the mixed layer and nutrient supply from depth (Zhou et al., 2020a). Spatially, Cai et al. (2015) found that POC export fluxes decreased with distance offshore in the northern SCS due to reduced POC stocks. Mesoscale processes can also promote POC export by pumping nutrient-replete waters from depth into the Ez (Zhou et al., 2013; 2020b). Regardless, POC export fluxes at different export horizons, and the sources of new nutrients that support export, remain understudied in the oligotrophic SCS.

In this study, we conducted sampling of [234]Th at high depth resolution in the Ez during the summer of 2017 to examine the structure of export productivity partitioning in the SCS basin. We calculated POC export fluxes based on [234]Th from both the NDL and Ez. Based on trap-derived masses and [15]N-isotopic balances, we estimated the relative contributions of different nutrient sources to export fluxes within the two-layer nutrient-based structure in the SCS at two stations with different hydrological features. Moreover, we related POC export fluxes from the two layers to their different biogeochemical forcings (especially the depth of the nutricline and the subsurface nutrient concentrations), to examine the controlling factors that potentially regulate POC export flux in the oligotrophic SCS.

## 2 Methods

### 2.1 Sample collection

Ship-based sampling occurred from June 5[th] to June 27[th], 2017 on the R/V Tan Kah Kee in the SCS basin (Fig 1) under the umbrella of the CHOICE-C II project (**Carbon cycle in the South China Sea: budget, controls and global implications, National Key Scientific Research Project**). Two mega stations (SEATS and SS1) and 9 normal stations were visited during the cruise. Typhoon Merbok passed by the SCS basin on June 10[th], 2017, before our field campaign. In order to examine the spatial variability of [234]Th, 4 stations (H01, H06, H08 and H11) clustered closely around station SS1 were sampled (Fig 1). Seawater samples were collected using 12-L or 10-L Niskin bottles attached to a Seabird 911 conductivity-temperature-depth (CTD) profiler.



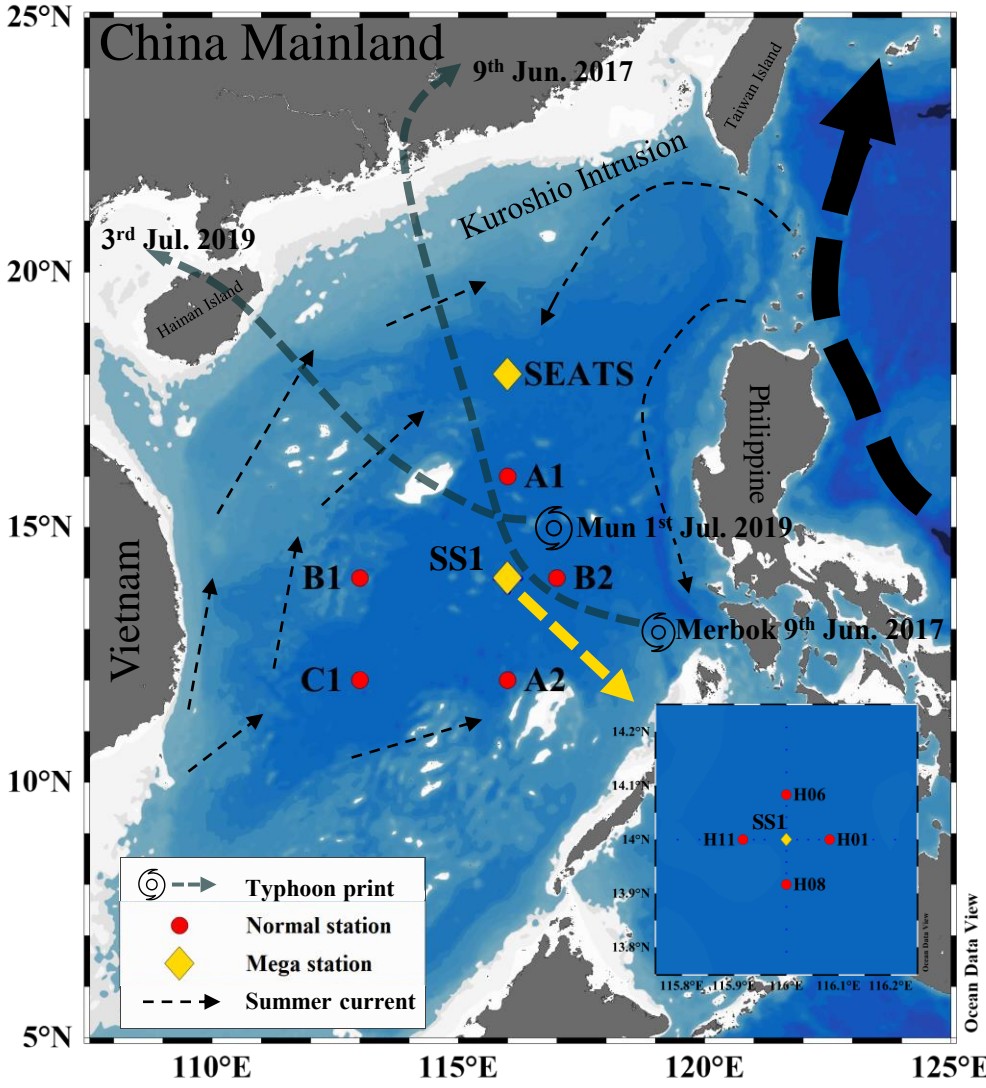

**Figure 1:** Map of the South China Sea (SCS) with sampling stations during June 2017. Yellow diamonds denote mega stations (SEATS and SS1) where high-resolution sampling was conducted at a 10-m interval in the euphotic zone; red circles denote normal stations where samples were collected at typical sampling depths of 5, 25, 50, 75 and 100 m. The general circulation pattern (adopted from Liu et al., 2016) is also shown. The dominant summer currents are also denoted by black dashed arrows. The dark blue dashed lines denote the paths of typhoon Merbok (sourced from the southeastern part of SCS on June 9th, 2017, adapted from Jiang et al., 2020) and Mun (sourced from the eastern part of the SCS on July 1st, 2019, adapted from Sun et al., 2021).



At the mega stations, high vertical resolution water samples were taken at a depth interval of 10 m within the Ez. For normal stations, lower resolution (5, 25, 50, 75 and 100 m) samples were collected. 4 L and 8 L seawater volumes were collected for total $^{234}$Th and particulate $^{234}$Th/POC analysis, respectively. Samples were collected using acid clean 4-L fluorinated bottles and filtered onto quartz microfiber (QMA) filters (25 mm diameter, 1.0 μm pore size). 500-mL of seawater was also collected for nutrient analysis from the Niskin bottles. Ancillary parameters, including potential temperature, salinity and fluorescence,

were accessed using a seabird CTD sensor. We calibrated the sensor-derived fluorescence with the Chl $a$ concentrations from discrete samples using the equation: Chl $a$ (mg m$^{-3}$) = 0.855×fluorescence (R$^2$ =0.87, n=139, Fig S1).

    In addition, we deployed an array of floating sediment traps for 72 hours at 50, 100 and 200 m at the SEATS station to collect sinking particles during the survey. The other mega station, SS1, was revisited during August 2019 for a 53-hour sediment trap deployment and sinking particle collection at the same depths after typhoon Mun passing by. At each depth of stations

SEATS and SS1, 12 cylindrical acrylic tubes (with a height of 50 cm and diameter of 10 cm) were assembled for different biogeochemical measurements. Before the deployment, the tubes were filled with prefiltered surface seawater and NaCl was added to supersaturation. After recovery, the tubes were placed under 4°C until the particles settled to the bottom. After removing the overlying supernatant, the particles were prefiltered through Nitex filters (120 μm pore size) to remove the visible zooplankton, and then collected on QMA filters (1.0 μm pore size) for elemental and isotopic analyses.

**2.2 $^{234}$Th analysis**

The small-volume (4 L) MnO$_2$ co-precipitation method was conducted for total $^{234}$Th analysis (Benitez-Nelson et al., 2001; Cai et al., 2006). The efficiency of thorium precipitation was monitored by $^{230}$Th. In detail, the seawater samples were acidified after collection and spiked with 200 μL of $^{230}$Th (17.38 dpm mL$^{-1}$). After an 8-hour period to allow equilibration between samples and tracers, the pH of seawater was raised to 8.05-8.20 using NH$_3$·H$_2$O before 0.375 ml KMnO$_4$ (3.0 g L$^{-1}$) and 0.20

ml MnCl$_2$ (8.0 g L$^{-1}$) were added. The MnO$_2$ precipitates collected for total $^{234}$Th and the particles filtered for particulate $^{234}$Th from the seawater samples on a QMA filter (25 mm, 1.0 μm) were dried in the oven overnight under 45°C. The filters were then packed with Teflon rings and discs (diameter of 23.5 cm, produced by RISØ National Laboratory, Denmark) covered by Al foil (density: 6.45 mg m$^{-2}$) and Mylar film. A gas-flow proportional low-level RISØ beta counter (Model GM-25-5) was used for $^{234}$Th counting. The first count was carried out immediately after the samples were set up, and the second count was

carried out after > 6 months for the background measurement. All $^{234}$Th samples were counted for 1000 minutes each time. The $^{230}$Th was monitored using $^{229}$Th, which was purified after iron precipitation and anion column exchange, and the concentrated finally diluted to 6 mL in 2% HNO$_3$. The samples were then settled into 15-ml centrifuge tubes and measured by inductively coupled plasma-mass spectrometry (ICP-MS) (Agilent 7700x). The average of all the recoveries was 88±12 % (mean±1σ, n = 97, range 73-98%). All $^{234}$Th data were recovery- and decay-corrected to the sampling time. The uncertainties

of $^{234}$Th data were propagated from the counting error, uncertainty from recovery and detection efficiency. The $^{238}$U activity was estimated by the following equation assuming conservative behavior with respect to salinity (Owens et al., 2011):





$$^{238}U = 0.0786 \times S - 0.314 \,, \tag{1}$$

### 2.3 POC, PN and δ$^{15}$N$_{PN}$ analyses

Following the measurement of particulate $^{234}$Th, the samples were carefully removed from the discs and placed in glass dishes.

Subsequently, the filters were dried at 50 ºC for 24 hours after adding 0.4 mL of HCl (1.0 μmol L$^{-1}$) to remove inorganic carbon. POC and PN concentrations were determined by an Elemental Analyzer- Isotope Ratio Mass Spectrometer (EA-IRMS) system (EA:vario PYRO cube and IRMS: Isoprime 100). At station SS1, we conducted 10 replicate POC samplings at 5, 100 and 200 m water depth to investigate the precision of bottle-collected POC. Our results show that standard deviations of our analyses were better than 13%, which agrees well with the result from the JGOFS cookbook (Knap et al., 1996). The errors

were included in the subsequent calculation of POC export fluxes. The particles from the sediment traps were treated the same as the suspended particles. The C and N contents and the isotopes of sinking particles were also analyzed by EA-IRMS.

### 2.4 $^{234}$Th scavenging model

The mass balance for $^{234}$Th in seawater can be described as Eq. (2) (Buesseler et al., 1992):

$$\frac{\partial A_{Th}^{total}}{\partial t} = \lambda(A_U - A_{Th}^{total}) - F_{Th}^{i} + F_{Th}^{i-1} + V \,, \tag{2}$$

where $F_{Th}^{i}$ is the $^{234}$Th scavenging flux in the layer $i$, and $F_{Th}^{i-1}$ is the $^{234}$Th scavenging flux above the layer $i$. We assume $F_{Th}^{i-1} = 0$ when we calculate $^{234}$Th flux for the initial layer ($i = 1$). $A_U$ and $A_{Th}^{total}$ are the $^{238}$U and total $^{234}$Th activities, and $\lambda$ is the $^{234}$Th decay constant (0.02876 d$^{-1}$); $V$, which is discussed below, is the term for physical effects, including advection and diffusion.

For particle export from the Ez, the deficit of total $^{234}$Th relative to $^{238}$U is integrated with depth to evaluate $^{234}$Th fluxes.

Under steady state (SS) ($\frac{\partial A_{Th}^{total}}{\partial t} = 0$), the $^{234}$Th export flux from the Ez ($F_{Th}^{Ez}$) is integrated by Eq. (3) (as shown in Fig 2):

$$\frac{\partial A_{Th}^{total}}{\partial t} = \lambda(A_U - A_{Th}^{total}) - F_{Th}^{i} + F_{Th}^{i-1} + V \,, \tag{3}$$

Similarly, $^{234}$Th export flux from the base of the NDL, $F_{Th}^{NDL}$, is estimated as follows:

$$F_{Th}^{NDL} = \int_{0}^{NDL} \left( A_U - A_{Th} \right) \times \lambda dz \,, \tag{4}$$



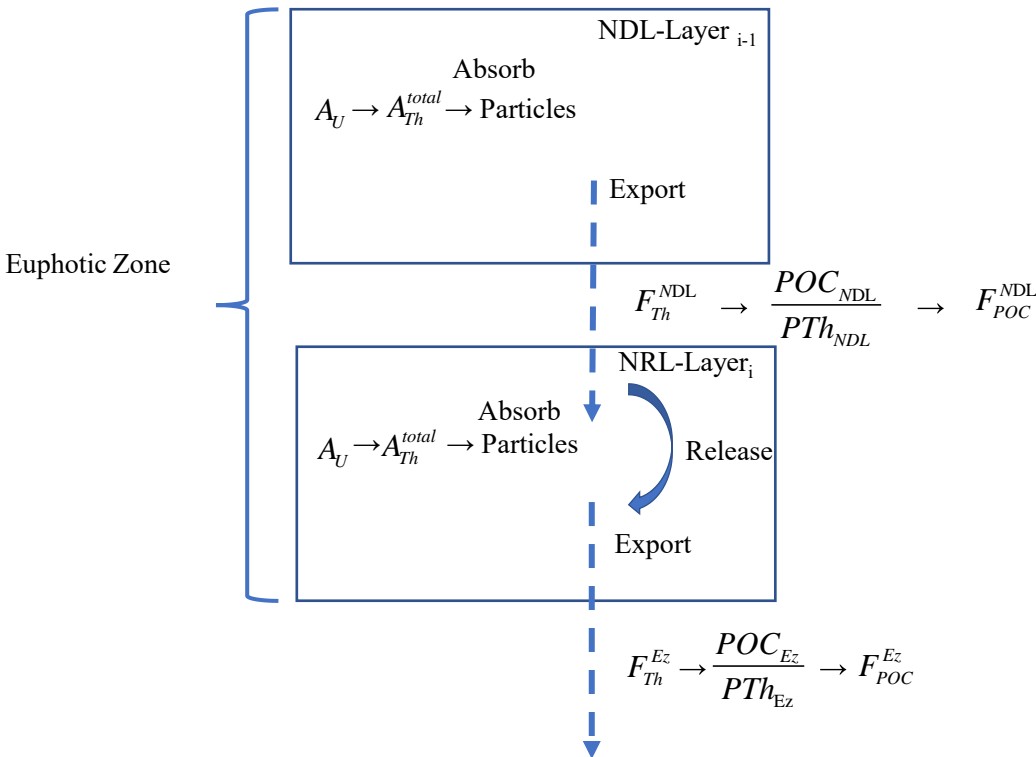

**Figure 2:** Schematic of the [234]Th model under the two-layer nutrient structure. All terms are defined in Equations (2)-(6).

### 2.5 POC export flux calculation

In this study, the [234]Th-derived POC export flux was calculated using the following equations:

$$F_{POC}^{Ez} = F_{Th}^{Ez} \times \frac{POC_{Ez}}{PTh_{Ez}} , \tag{5}$$

where $F_{Th}^{Ez}$, $\frac{POC_{Ez}}{PTh_{Ez}}$ and $F_{POC}^{Ez}$ are the [234]Th flux, particulate POC/[234]Th ratio, and POC flux at the Ez base, respectively.

$$F_{POC}^{NDL} = F_{Th}^{NDL} \times \frac{POC_{NDL}}{PTh_{NDL}} , \tag{6}$$

where $F_{Th}^{NDL}$, $\frac{POC_{NDL}}{PTh_{NDL}}$ and $F_{POC}^{NDL}$ are the [234]Th flux, particulate POC/[234]Th ratio, and POC export flux at the NDL base, respectively.

Sediment trap-derived POC export fluxes were calculated as follows:



$$F_{POC-Trap} = \frac{POC_{Measured}}{\Delta t \times A_{TrapTube}}, \tag{7}$$

where POC is the concentration of organic carbon on the particles collected by the traps, $\Delta t$ is the duration of trap deployments, and $A_{TrapTube}$ is the area of the trap tube.

### 2.6 The depth of the euphotic zone

The euphotic zone depth (Zeu or the Ez base, in m) is defined optically, based on Wu et al. (2021), as the depth where the usable solar radiation (USR) equals 0.9% of the surface USR, which is close to the depth where the photosynthetic available

radiation (PAR) equals 0.5% of the PAR value at the sea surface. *In-situ* Zeu during the cruise was obtained from profiling PAR data recorded by the optical sensor (Biospherical QCP2300-HP) on the CTD.

### 2.7 Nutrient analysis and nutricline depth

Nutrients were analysed onboard using a Four-channel Continuous Flow Technicon AA3 Auto-Analyzer (Bran-Lube GmbH). The detection limits for both N+N (nitrate plus nitrite, termed as dissolved inorganic nitrogen, DIN) and SRP (soluble reactive

phosphate) were 0.03 μmol L$^{-1}$. The top of the nutricline in this study was defined as the depth at which the DIN concentration reached 0.1 μmol L$^{-1}$ (Dore and Karl, 1996; Winn et al., 1996). The layers above, and below to the base of Ez, were defined as the as NDL and NRL, respectively.

## 3 Results

### 3.1 Environmental settings

The potential temperature-salinity ($\theta$-S) diagram of the water column reveals distinctive hydrological features between stations in the SCS basin. As shown in Figure 3, the potential temperature reached around 30 °C at the sea surface and decreased with depth, while the salinity increased with depth in the upper 250 m before reaching its peak in the subsurface and then decreased to approximately 34.5 at deeper depths. The spatial differences in the hydrography were significant (Fig 4). The surface mixed layer depth (MLD, defined as the depth where the potential density $\sigma_\theta$ increased by 0.03 kg m$^{-3}$ compared to the value at the

sea surface, Cornec et al., 2021) at stations SEATS, A1, A2 and C1 was shallower (20-39 m) than at other stations in the sampling region (MDL>40 m, Table 1, Fig 4). The shallower MLD and isoclines (i.e., thermocline and halocline) might indicate upward displacement of water at those stations. Du et al. (2021) attributed such vertical shift of the isoclines to mesoscale processes or basin scale circulation. Indeed, most of these stations (SEATS, A1 and C1) were under the influence of eddies during the sampling periods as revealed by the Sea level Anomaly (SLA) map (Fig S2); modeling indicates Stations





C1 and A2 were impacted by cold water sourced from the southwestern SCS basin when they were sampled (Fig S3) (Gan et al., 2016).

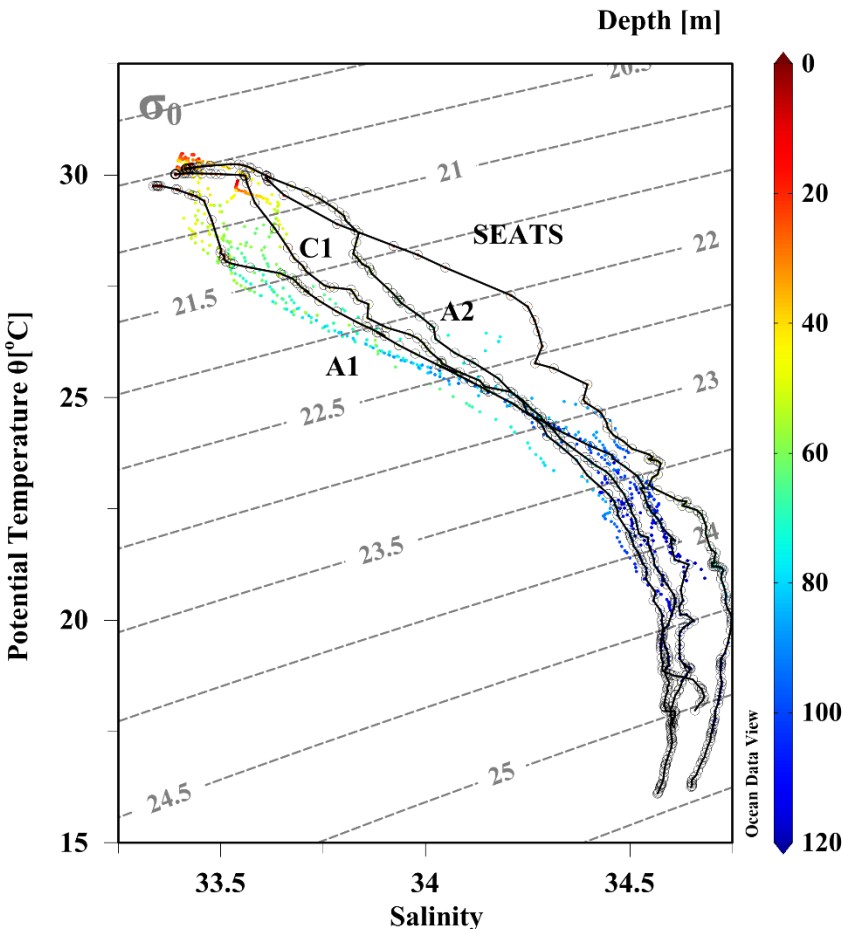

**Figure 3:** Plot of potential temperature (θ) vs. salinity (S) (θ-S diagram) for the sampling stations in the South China Sea basin during June 2017. Superimposed are sampling depths as shown in the color bar. The stations with shallow mixed layer depths (MLDs), i.e., stations SEATS, A2, C1 and A1, are highlighted with solid black lines.







**Figure 4:** Vertical profiles of temperature (a), salinity (b), dissolve inorganic nitrogen (nitrate + nitrite, DIN, c) and Chl *a* (d). The MLD (red dash), interpolated depth of DIN=0.1 μmol L⁻¹ (top of nutricline, yellow dash) and subsurface Chl *a* Maximum (SCM, green dash) are also shown.





**Table 1:** Surface mixed layer depths (MLDs), export horizon depths, 1D- steady state $^{234}$Th fluxes, POC/$^{234}$Th ratios, and POC

fluxes at stations in the upper oligotrophic South China Sea basin during June 2017

| Station | [1]MLD | [2]NDL base | [3]Ez base | $^{234}$Th flux@NDL | $^{234}$Th flux @Ez | POC/$^{234}$Th ratio@NDL | POC/$^{234}$Th @Ez | POC export flux @ NDL | POC export flux @ Ez |
|---------|--------|-------------|------------|---------------------|---------------------|--------------------------|--------------------|------------------------|----------------------|
| | [m] | [m] | [m] | dpm m$^{-2}$ d$^{-1}$ | dpm m$^{-2}$ d$^{-1}$ | µmol C dpm$^{-1}$ | µmol C dpm$^{-1}$ | mmol C m$^{-2}$ d$^{-1}$ | mmol C m$^{-2}$ d$^{-1}$ |
| SEATS | 27 | 50 | 80 | 362±34 | 522±43 | 4.4±0.6 | 5.5±0.7 | 1.6±0.6 | 2.9±0.7 |
| C1 | 36 | 59 | 87 | 598±57 | 602±22 | 6.2±0.8 | 2.9±0.4 | 3.7±0.9 | 1.7±0.4 |
| A1 | 27 | 57 | 88 | 603±98 | 585±100 | 7.1±0.9 | 5.2±0.7 | 4.3±1.2 | 3.0±0.8 |
| A2 | 39 | 63 | 96 | 624±52 | 839±59 | 6.3±0.8 | 2.7±0.3 | 3.9±0.9 | 2.2±0.4 |
| B2 | 44 | 71 | 102 | 204±57 | 267±69 | 8.2±1.1 | 8.3±1.1 | 1.7±1.2 | 2.2±1.2 |
| SS1 | 43 | 81 | 111 | 613±42 | 631±48 | 4.0±0.5 | 3.1±0.4 | 2.4±0.5 | 2.0±0.4 |
| B1 | 50 | 78 | 87 | 361±63 | 421±64 | 4.1±0.5 | 3.8±0.5 | 1.5±0.6 | 1.6±0.6 |
| H08 | 42 | 80 | 106 | 376±61 | 462±68 | 5.8±0.8 | 4.6±0.6 | 2.2±0.8 | 2.1±0.7 |
| H11 | 48 | 82 | 106 | 360±61 | 393±66 | 3.2±0.4 | 4.1±0.5 | 1.1±0.5 | 1.6±0.6 |
| H06 | 52 | 87 | 115 | 439±63 | 462±66 | 3.2±0.4 | 2.8±0.4 | 1.4±0.5 | 1.3±0.4 |
| H01 | 48 | 99 | 107 | 351±70 | 350±70 | 3.3±0.4 | 3.3±0.4 | 1.2±0.5 | 1.2±0.5 |

[1]The MLD was defined as the depth where the potential density $\sigma_\theta$ increased by 0.03 kg m$^{-3}$ compared to values at sea surface

(Cornec et al., 2021).

[2]The NDL base, namely the top of the nutricline, was interpolated to the depth where DIN = 0.1 µmol L$^{-1}$ based on the DIN

distribution near the SCM.

[3]The Ez base was estimated to be the depth where PAR is 0.5% of the PAR value at the sea surface.





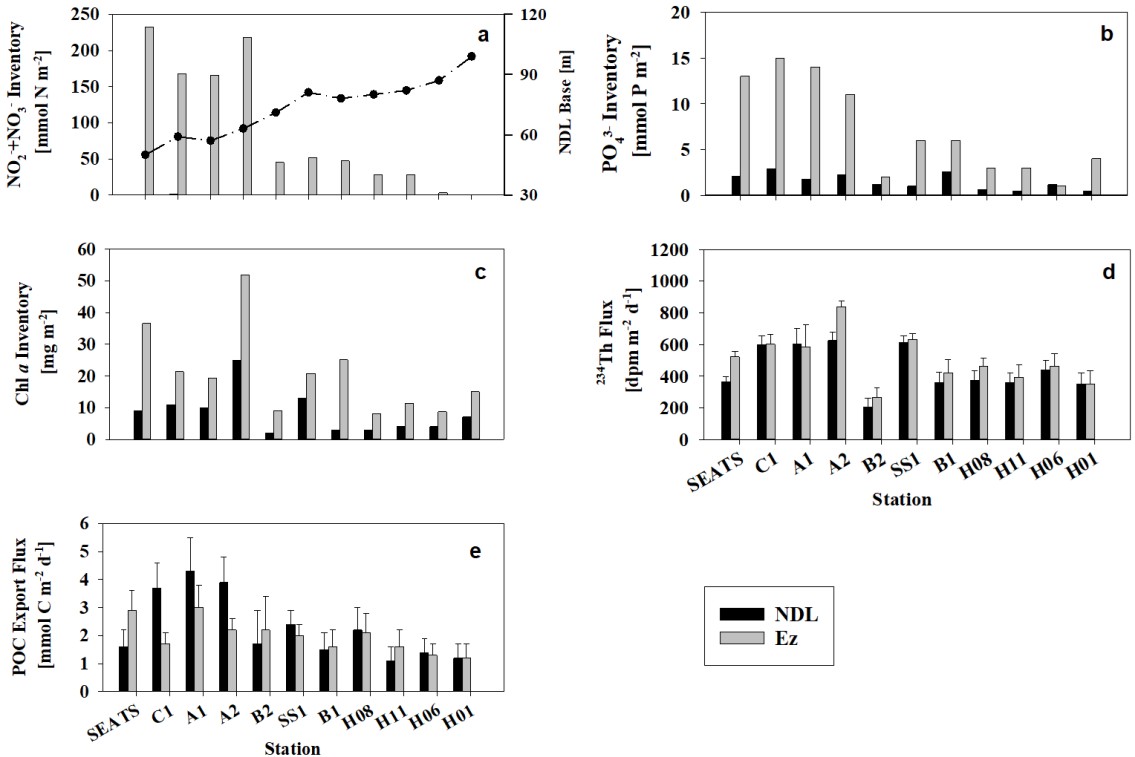

**Figure 5:** Integrated inventories of DIN (a) and DIP (b) in both the NDL (black) and Ez (grey). Also shown are the partitioned Chl $a$ stocks (c), integrated partitioned $^{234}$Th fluxes (d) and $^{234}$Th-derived POC export fluxes (e). The high and low nutrient inventories correspond to shallow and deep nutriclines (a) (dotted line, NDL base), respectively.


Chl $a$ concentrations at the 4 stations with shallower nutriclines were enhanced in response to elevated nutrient levels resulting in shallower depths of subsurface Chl $a$ maxima (SCM, Fig 4d) relative to other stations, (55-80 m vs 85-108 m). Chl $a$ inventories at stations with high nutrient inventories (23.6-52.2 mg m$^{-2}$, average 29.6±4.8 mg m$^{-2}$) were significantly higher ($p < 0.05$) than at others stations (8.0-22.8 mg m$^{-2}$, average 14.0±4.6 mg m$^{-2}$, Fig 5c).

**3.2 $^{234}$Th and POC variability**

The variability of total $^{234}$Th and Chl $a$ versus depth is shown in Fig 6. The activities of total $^{234}$Th ranged from 1.70±0.05 to 2.73±0.05 dpm L$^{-1}$, with an average of 2.30±0.31 dpm L$^{-1}$ (n = 97, Fig 6), and all stations displayed similar patterns. Generally, $^{234}$Th was deficit relative to $^{238}$U in the upper Ez, and was in equilibrium or excess at the base of and/or below Ez. The $^{234}$Th deficit peaked within the NDL and largely diminished in the NRL, implying a large amount of particle removal occurred in the NDL but low export or high remineralization in NRL. The $^{234}$Th activity minimum (1.70±0.17 dpm L$^{-1}$) appeared at a depth





of 25 m at station A1 (one of stations characterized by a shallow MLD and nutricline). $^{234}$Th activity at the stations surrounding station SS1 showed little spatial variability: the differences in $^{234}$Th activity were less than 0.1 dpm L$^{-1}$ at the same depth.

**Figure 6:** Depth profiles of DIN (blue open circle, μmol L$^{-1}$), total $^{234}$Th activity (black dot, dpm L$^{-1}$), $^{238}$U activity (black dash, dpm L$^{-1}$) and Chl *a* concentration (grey line, mg m$^{-3}$) in the South China Sea basin. The defined export horizons of the NDL base (blue dash) and Ez base (yellow dash) are also shown. The deficit of $^{234}$Th relative to $^{238}$U was the most pronounced in the province where DIN was too low to be detected.



Particulate $^{234}$Th ranged from 0.13±0.01 dpm L$^{-1}$ to 0.47±0.01 dpm L$^{-1}$ (with an average of 0.25±0.11 dpm L$^{-1}$, n=83) (Fig

7). At most stations the profiles of particulate $^{234}$Th shared similar depth patterns with Chl $a$, with the maximum values

appearing in the subsurface water, while at stations H01 and H06, particulate $^{234}$Th generally increased with depth in the upper

100 m, and showed little station to station variability. The maximum of particulate $^{234}$Th appearing at both surface and

subsurface at station B2 suggested complicated biogeochemistry of $^{234}$Th on particles.

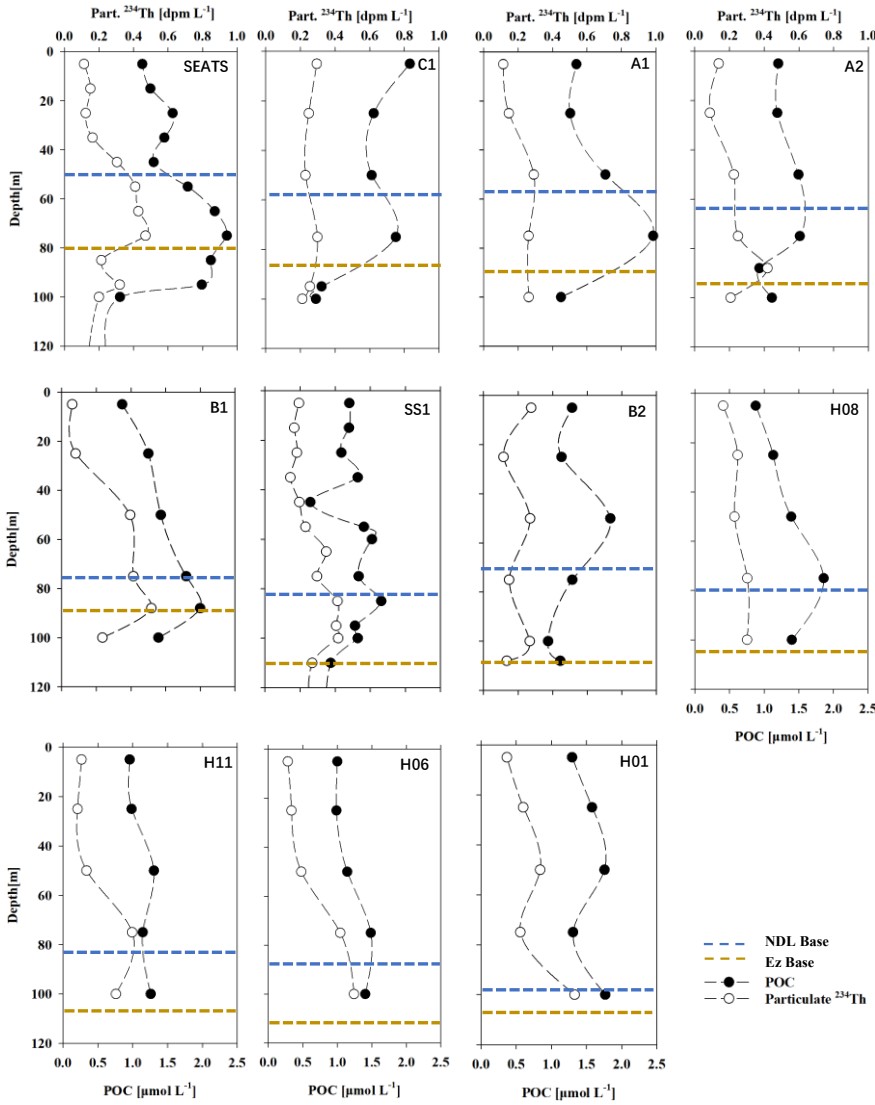

**Figure 7:** Profiles of POC (black dots, μmol L$^{-1}$) and particulate $^{234}$Th activity (PTh, open circles, dpm L$^{-1}$) at all stations

sampled in the South China Sea basin in June 2017. The bases of both the NDL (blue dashed line) and Ez (yellow dashed line)

are also shown.





POC concentrations ranged from 0.83 µmol L$^{-1}$ to 2.5 µmol L$^{-1}$ (with an average of 1.2±0.44 µmol L$^{-1}$, n=83) (Fig 7). At most stations, the POC concentration was low (average 1.1±0.2 µmol L$^{-1}$) in surface water and generally increased with depth until reached its maximum at the SCM layer, and then decreased again with depth. However, at some stations (C1, B2), there were

POC peaks appearing in both the surface water and the SCM layer.

### 3.3 1D SS water column-integrated and sediment trap-derived $^{234}$Th flux

Calculated $^{234}$Th fluxes at different export horizons (i.e., NDL and Ez base) are shown in both Table 1 and Fig 5d. $^{234}$Th fluxes at the Ez base mostly ranged from 267±69 to 839±59 dpm m$^{-2}$ d$^{-1}$. $^{234}$Th fluxes at NDL base ranged from 204±57 to 624±52 dpm m$^{-2}$ d$^{-1}$, which accounts for 69-100% of $^{234}$Th fluxes at the Ez base.

$^{234}$Th fluxes at the Ez base in this study were within the ranges (62-1365 dpm m$^{-2}$ d$^{-1}$) found in prior studies in the SCS basin (Cai et al., 2008; Cai et al., 2015; Zhou et al., 2013; Zhou et al., 2020a). Although $^{234}$Th fluxes at the base of the NDL have rarely been quantified, the particle-scavenging rate at any export horizon can be determined using $^{234}$Th methodology (Buesseler et al., 2020b). $^{234}$Th fluxes at the NDL base in the oligotrophic SCS (77-942, averaged 349±296 dpm m$^{-2}$ d$^{-1}$) showed little difference from the integrated results of prior studies (Cai et al., 2015; Zhou et al., 2020a) due to the similarity

of measured $^{234}$Th activities in the NDL.

Sediment trap-derived $^{234}$Th fluxes at SEATS were 589±2 dpm m$^{-2}$ d$^{-1}$ at the NDL base (50 m), representing over 50% of the $^{234}$Th flux, and 830±2 dpm m$^{-2}$ d$^{-1}$ near the Ez base (100 m). The trap-derived $^{234}$Th fluxes were higher than, but within 2-fold, of the fluxes derived from bottle sampled $^{234}$Th (362±34 dpm m$^{-2}$ d$^{-1}$ at 50 m and 471±46 dpm m$^{-2}$ d$^{-1}$ at 100 m) at both export horizons. Nevertheless, the partitioning between NDL and Ez particle fluxes, based on both techniques, is similar, which

further supports that our $^{234}$Th-$^{238}$U disequilibrium-based fluxes are representative.

### 3.4 POC/$^{234}$Th ratios based on bottle filtration and sediment traps

Bottle-derived POC/$^{234}$Th profiles in the Ez are shown in Fig 8. They range from 2.6 to 15.7 µmol dpm$^{-1}$ (with an average of 5.6±3.3 µmol dpm$^{-1}$, N=83), peak in the upper 25 m and generally decrease with depth at all stations. POC/$^{234}$Th differences between most stations gradually diminished with depth and converged near 4.2±1.6 µmol dpm$^{-1}$ at the base of the Ez. The

decreasing pattern of POC/$^{234}$Th was not observed at stations SEATS and H11 (as noted in Fig 8a).

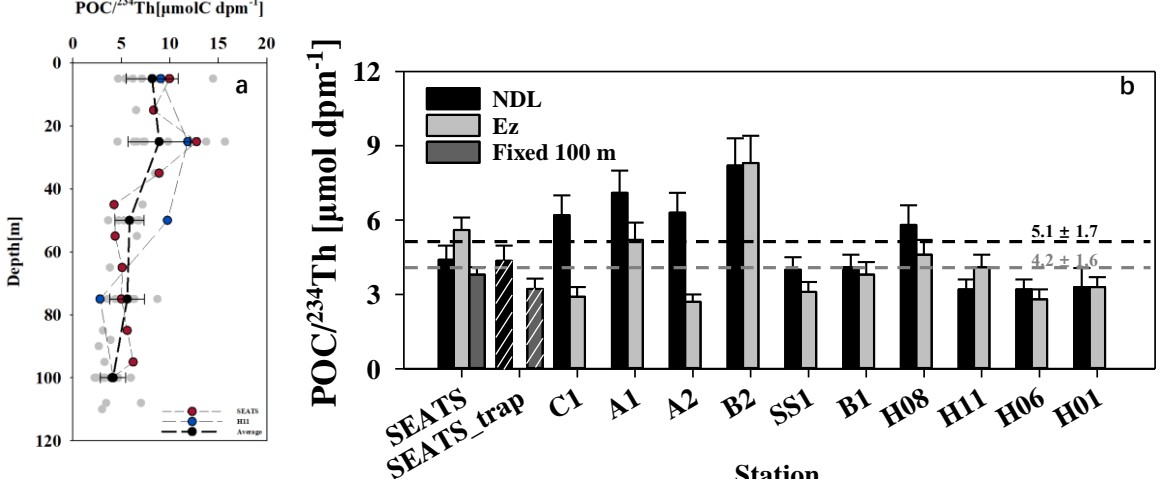

**Figure 8:** Water-column POC/$^{234}$Th ratios from bottle filtration, with the averages (black dots with dashed line) at each sampling depth plotted against depth (a). Also shown are the bottle- and trap-derived POC/$^{234}$Th ratios (bar with white stripes) at the bottom of the NDL (black), base of the euphotic zone (light grey), and fixed 100 m depth (dark grey) (b). Generally, the

variability of POC/$^{234}$Th decreased as depth increased and converged around 4.2±1.6μmol dpm$^{-1}$ at the Ez base. No significant variability (within 2-fold) was found between POC/$^{234}$Th ratios derived from bottle and trap samples accessed at the same sampling depths at station SEATS.

POC/$^{234}$Th from sediment traps was only measured at station SEATS, and were 4.7 and 3.2 μmol dpm$^{-1}$ at 50 and 100 m,

respectively (Fig.8b). These values are comparable with the bottle-derived POC/$^{234}$Th ratios observed in this study at similar depths at the same site.

### 3.5 $\delta^{15}N_{PN}$ from sediment traps

The $\delta^{15}N_{PN}$ values for the trap samples varied between 2.6‰ to 6.7‰ in the upper 200 m at station SEATS and SS1, showing an increasing trend with depth. Specifically, the lightest $\delta^{15}N_{PN}$ of 2.6 ‰ was observed at 50 m within the NDL, and the $\delta^{15}N_{PN}$

increased to 4.7‰ at the Ez base (about 100 m). Below the Ez, the $\delta^{15}N_{PN}$ value increased to 6.7‰ at 200 m at station SEATS. A similar pattern of $\delta^{15}N_{PN}$ was also found at station SS1, with the lightest $\delta^{15}N_{PN}$ of 4.1 ‰ at 50 m, an intermediate value of 5.80 ‰ at 100 m and the heaviest value of 6.00 ‰ at 200 m. The observed $\delta^{15}N_{PN}$ values at both stations were comparable to previous results (3.3-7.3 ‰) from sinking particles collected by sediment traps in the upper 500 m around SEATS (Kao et al., 2012; Yang et al., 2017). Yang et al. (2017) found a $\delta^{15}N_{PN}$ value of 4.9‰ at 100 m at station SEATS, which was very consistent

with our observation at the same depth. These results suggest that inter-annual variations in $\delta^{15}N_{PN}$ from the upper ocean in the





SCS may be limited, and the $\delta^{15}N_{PN}$ value at station SS1 from the cruise in 2019 could be comparable to that in this campaign. We thus diagnose the nutrient sources of sinking particles at stations with different environmental settings without focusing on temporal variability.

## 4 Discussion

**4.1 Partitioning of $^{234}$Th fluxes within the euphotic zone in the oligotrophic SCS**

### 4.1.1 Impacts of physical transport on $^{234}$Th flux estimation

To estimate the $^{234}$Th flux induced by the particle scavenging, both horizontal and vertical transports of $^{234}$Th need to be evaluated.

In this study, the physical transport is estimated as follows:

$$V = u \times \frac{\partial A_{Th}}{\partial x} + v \times \frac{\partial A_{Th}}{\partial y} + w \times \frac{\partial A_{Th}}{\partial z} + K_x \frac{\partial^2 A_{Th}}{\partial x^2} + K_y \frac{\partial^2 A_{Th}}{\partial y^2} + K_z \frac{\partial^2 A_{Th}}{\partial z^2} \tag{8}$$

where $u$, $v$, and $w$ are the zonal, meridional, and upwelling velocities respectively, $\frac{\partial A_{Th}}{\partial x}$, $\frac{\partial A_{Th}}{\partial y}$ and $\frac{\partial A_{Th}}{\partial z}$ are $^{234}$Th activity gradients from west to east, south to north, and upward. $K_x, K_y$ and $K_z$ are diffusivities from west to east, south to north, and upward, respectively, and $\frac{\partial^2 A_{Th}}{\partial x^2}$, $\frac{\partial^2 A_{Th}}{\partial y^2}$ and $\frac{\partial^2 A_{Th}}{\partial z^2}$ are the second derivatives of $^{234}$Th activity distributions.

Due to the lack of *in situ* vertical current velocity ($w$) during the cruise, the climatological $w$ and $K_z$ from modeling results 305 (Gan et al., 2016) were applied to the equation to estimate the impacts of vertical advection and diffusion on the $^{234}$Th flux. The integrated vertical transport flux was 43 dpm m$^{-2}$ d$^{-1}$ and can be considered negligible (less than 10%) compared to the vertical scavenging flux at station SS1. This is consistent with Cai et al. (2008) who also stated that the vertical term could be neglected for $^{234}$Th flux estimation in the SCS basin.

The apparent diffusivity around station SS1 is estimated as $\sim 4 \times 10^5$ cm$^2$ s$^{-1}$ (Okubo, 1971) from empirically derived oceanic 310 diffusion diagrams, and we simplified the horizontal diffusive term in Eq. (8) based on Benitez-Nelson et al. (2000):

$$V_{diffusion} = \sqrt{\frac{\left[K_x \left(A_{Th-H11} - 2 \times A_{Th-SS1} + A_{Th-H01}\right)\right]^2}{\Delta x^2} + \frac{\left[K_y \left(A_{Th-H08} - 2 \times A_{Th-SS1} + A_{Th-H06}\right)\right]^2}{\Delta y^2}} \tag{9}$$

Thus, the $^{234}$Th flux derived from horizontal diffusion was considerably low (approximately 0.1 dpm m$^{-2}$ d$^{-1}$). The highly variable *in situ* horizontal current velocities at station SS1 from the Acoustic Doppler Current Profiler (ADCP) showed a wide range from 0.01 m s$^{-1}$ to 0.3 m s$^{-1}$ in the upper 200 m. As those current velocities were measured instantaneously and their time 315 scales did not match those for $^{234}$Th ($\sim$ 20 days), we thus applied model-derived time-integrated data (three-month average) to





the equation instead. The model-derived $u$ and $v$ ranged from 0.007 m s$^{-1}$ to 0.2 m s$^{-1}$ in the upper 100 m. Based on those velocities, the $^{234}$Th flux from horizontal transport (96 dpm m$^{-2}$ d$^{-1}$) was <20% of the $^{234}$Th flux using the SS model in the upper 100 m (634±46 dpm m$^{-2}$ d$^{-1}$), similar to prior studies in oligotrophic ecosystems (Buesseler et al., 2020b). Therefore, a 1D-model assumption is applicable in this study for $^{234}$Th flux estimation.

**4.1.2 $^{234}$Th fluxes at the NDL and Ez bases**

Assuming a 1D-SS model is valid in the case of low particle fluxes ($^{234}$Th flux < 800 dpm m$^{-2}$ d$^{-1}$, Savoye et al., 2006), the partitioned particle flux at the NDL base was comparable (88±11%) to that at the Ez base. This vertical structure indicates that the NDL base should be a hotspot for particle scavenging. The trap-derived $^{234}$Th fluxes (589±2 dpm m$^{-2}$ d$^{-1}$ at 50 m and 830±2 dpm m$^{-2}$ d$^{-1}$ at 100 m) were slightly higher compared to bottle-derived $^{234}$Th fluxes (362±34 dpm m$^{-2}$ d$^{-1}$ at 50 m and 471±46

dpm m$^{-2}$ d$^{-1}$ at 100 m). The higher trap-derived $^{234}$Th fluxes might possibly be related to incomplete removal of zooplankton (Buesseler et al., 2020b). In addition, the inconsistency between the two methods could be due to the different time scales (Umhau et al., 2019). Regardless of the differences in $^{234}$Th flux estimations from the separate methods, the similar vertical partitioning from the both bottle- and trap-derived $^{234}$Th fluxes indicated substantial particle scavenging at the bases of both the NDL and Ez in the oligotrophic SCS.

It is interesting to note that at stations with higher nutrient inventories, $^{234}$Th fluxes (362±34-624±52 dpm m$^{-2}$ d$^{-1}$, average 547±107 dpm m$^{-2}$ d$^{-1}$ at the NDL base and 522±45-839±59 dpm m$^{-2}$ d$^{-1}$, average 637±120 dpm m$^{-2}$ d$^{-1}$ at the Ez base) are significantly higher (by approximately 100-200 dpm m$^{-2}$ d$^{-1}$) than those at other stations ( 210±38-520±31 dpm m$^{-2}$ d$^{-1}$, average 359±90 dpm m$^{-2}$ d$^{-1}$ at the NDL base, and 204±57-613±42 dpm m$^{-2}$ d$^{-1}$, average 427±105 dpm m$^{-2}$ d$^{-1}$ at the Ez base, Fig 5c). This regional pattern of $^{234}$Th fluxes might result from differences in nutrient distributions, as $^{234}$Th has thus far been an

indispensable tool to trace biogenic particle scavenging (Ceballos-Romero et al., 2022 and references therein). Whether these high and low $^{234}$Th fluxes would respectively drive similar POC export fluxes at stations with high and low nutrient inventories remains to be determined.

**4.2 POC/$^{234}$Th ratio and $^{234}$Th-derived POC fluxes in the SCS basin**

**4.2.1 Variability in bottle- and trap-derived POC/$^{234}$Th ratios**

Determining POC/$^{234}$Th ratios on particles at the export horizons is essential for converting $^{234}$Th fluxes to POC export fluxes. POC/$^{234}$Th ratios can, however, vary three orders of magnitude between different regions, depths, seasons and even particle sizes (Buesseler et al., 2006; Puigcorbé et al., 2020). The variability in POC/$^{234}$Th is possibly due to the combined effect of particle generation, aggregation, remineralization, and particulate $^{234}$Th decay (Cai et al., 2006). As shown in Fig 7a, water-column POC/$^{234}$Th ratios decreased gradually with depth and varied within 5 μmol dpm$^{-1}$ below the 50 m. This decreasing

tendency of POC/$^{234}$Th ratios was highly consistent with results from prior studies conducted in tropical-subtropical oligotrophic ecosystems despite differing sampling devices (Puigcorbé et al., 2020). The maximum ratio with the highest





variability was observed in the upper 25 m, at a depth where primary production usually peaks in oligotrophic ecosystems (Xie et al., 2018; Buesseler et al., 2020b). Even though POC/$^{234}$Th ratios determined from bottle filtration are variable in prior studies, they are strongly coupled to ratios from sediment traps (Gustafsson et al., 2013), which are considered to represent the
ratio on sinking particles. POC/$^{234}$Th ratios based on bottle filtration and sediment traps in this study were also compared to each other at the same depth at station SEATS: The POC/$^{234}$Th ratios were 4.2 and 3.2 µmol dpm$^{-1}$ on trap samples at 50 and 100 m, similar to bottle-filtration derived POC/$^{234}$Th ratios (4.4±0.6 and 3.8±0.6 µmol dpm$^{-1}$ at 55 and 100 m, respectively). Besides bottle- and trap-derived POC/$^{234}$Th ratios, the POC/$^{234}$Th ratio on large-size particles (> 53 µm and assumed to be sinking particles, Buesseler et al., 2006) retrieved from *in situ* pumping also decreased with depth at station SEATS (Cai et al.,
2006) and converged to a narrow range of 1.8-4.1 µmol dpm$^{-1}$ at 100 m in the SCS basin (Chen, 2008). Due to a lack of trap or pump deployment at all sites, and considering the similarity of POC/$^{234}$Th ratios using different methodologies, POC/$^{234}$Th ratios based on bottle filtration were used for POC flux estimation.

POC/$^{234}$Th ratios at the Ez base varied from 2.8±0.4 to 8.3±0.7 µmol dpm$^{-1}$ (with an average of 4.2±1.6 µmol dpm$^{-1}$, Fig 8b), which is comparable with previously published results (e.g., 1.6-5.3 µmol dpm$^{-1}$ with an average of 4.2±l.6 µmol dpm$^{-1}$) from
the SCS basin (Cai et al., 2015; Zhou et al., 2013; Zhou et al., 2020a). POC/$^{234}$Th ratios at the NDL base were generally higher than those at Ez base, ranging from 3.2±0.4 to 8.2±1.1 µmol dpm$^{-1}$ (with an average of 5.1±1.7 µmol dpm$^{-1}$).

We found variability in POC/$^{234}$Th ratios was insignificant between stations with shallow and deep nutriclines: The POC/$^{234}$Th ratio at the NDL base ranged from 4.4±0.6-7.1±0.9, average 6.0±1.0, µmol dpm$^{-1}$ at stations with shallow nutriclines (i.e., Sta. SEATS, C1, A1 and A2), which is slightly higher than the values at other sites (range 3.2±0.4-8.2±1.1, average at 4.5±1.7
µmol dpm$^{-1}$). On the other hand, the POC/$^{234}$Th ratios at the Ez base ranged from 2.9±0.4-5.5±0.7, averaged 4.0±1.3 µmol dpm$^{-1}$ at stations with shallow nutriclines, which was similar to the POC/$^{234}$Th ratios at other sites (range 2.8±0.4-8.3±1.1, average 4.3±1.7 µmol dpm$^{-1}$). The relatively low POC/$^{234}$Th at the NDL base at stations with deep nutriclines may be explained by higher particle remineralization rates with increasing depth. Based on similar ranges of $^{234}$Th fluxes and POC/$^{234}$Th ratios, the estimated POC export fluxes in this study were consistent with prior studies in the SCS basin (Cai et al., 2015; Zhou et al.,
2020a).

**4.2.2 POC export fluxes at different export horizons**

POC export fluxes were estimated after combining the partitioned $^{234}$Th fluxes and POC/$^{234}$Th ratios. $^{234}$Th-derived POC export fluxes ranged from 1.2±0.5 to 3.0±0.8 mmol C m$^{-2}$ d$^{-1}$ at the base of the Ez, and from 1.2±0.6 to 4.3±1.2 mmol C m$^{-2}$ d$^{-1}$ at the base of the NDL (Fig 5e and Table 1). POC export fluxes estimated in this study are of the same order of magnitude as previous
estimates in the SCS basin (Zhou et al., 2013; 2020a; Cai et a., 2015).

To assess the POC export flux using different methods, we compared $^{234}$Th- and trap-derived POC export fluxes at station SEATS. POC export fluxes were comparable near the Ez base (2.9±0.9 and 2.7±0.3 mmol C m$^{-2}$ d$^{-1}$ for $^{234}$Th- and trap-derived, respectively). However, the $^{234}$Th-derived POC export flux of 1.6±0.6 mmol C m$^{-2}$ d$^{-1}$ was slightly lower than the trap-derived POC export flux (2.8±0.3 mmol C m$^{-2}$ d$^{-1}$) at 50 m at station SEATS. The lower $^{234}$Th-derived POC export flux at 50 m may



indicate potential contamination by organics in the traps (e.g., swimmers) that would result in higher measured POC fluxes in the oligotrophic SCS basin. A recent study of the EXPORTS program found that swimmers could increase the measured POC export flux by 2-fold in the traps (Estapa et al., 2021). Although slight disagreement between different methods was often noted and difficult to assign causes (Hung and Gong, 2007; Stewart et al., 2007; Lampitt et al., 2008; Haskell II et al., 2013; Buesseler et al., 2020b), we clearly found substantial POC export fluxes at the NDL base that were comparable to those at the

Ez base in the SCS. A recent study based on $^{234}$Th and sediment traps in the oligotrophic Gulf of Mexico also found particle production dominates in the upper Ez (0-60 m) where nutrients are depleted (Stukel et al., 2021). The results above conflict with previous knowledge suggesting that POC export flux from the nutrient-depleted mixed layer is extremely low (Coale and Bruland, 1987). The substantial POC export flux at the NDL base was highly correlated to the Chl $a$ inventory, an index of biomass in the corresponding layer (Fig 9a). In this regard, the sources of new nutrients that support the relatively high biomass

in the NDL and drive the POC export fluxes at the NDL base in the SCS basin need to be constrained.



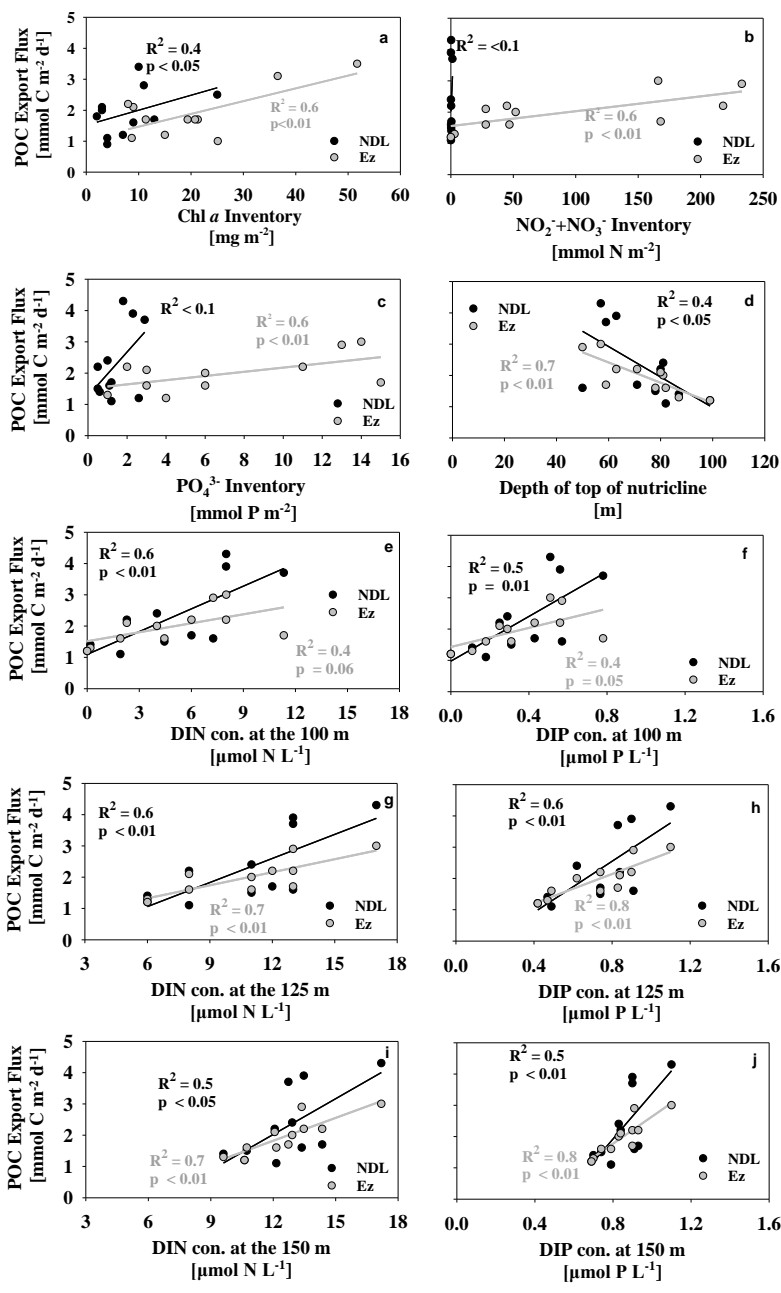

**Figure 9:** Relationship between POC export flux at the NDL base (black dots) and Ez base (grey dots) vs. Chl *a* (a), DIN (b) and DIP inventories (c) in the corresponding layers. Also plotted are the relationships between the depth of the top of the nutricline (d), and DIN and DIP concentrations in subsurface water at 100, 125 and 150 m versus partitioning POC export fluxes (e-j).





### 4.3 Diagnosis of nutrient sources supporting particle export in the oligotrophic SCS

#### 4.3.1 Correlation between POC export flux and subsurface nutrient concentrations

To diagnose the nutrient sources that support substantial POC export fluxes at different export horizons, we examined the
relationship between partitioned POC export fluxes and nutrient inventories in corresponding layers. Nutrient stocks might
regulate POC export fluxes at the Ez base based on their positive correlation (Fig 9b & c). However, a relatively poor
relationship between POC export fluxes at the NDL base and nutrient inventories in the NDL was found, which suggests that
the *in situ* nutrients in the NDL interior are insufficient to support the POC export from this horizon.

Since *in situ* nutrients in the NDL were insufficient to support POC export flux at the NDL, other external nutrient sources
likely influenced POC export flux in the nutrient-deleted ecosystems. Episodic events (e.g., eddies and typhoons) that can
transport subsurface nutrients into nutrient-deficient regimes have been confirmed in other oligotrophic ocean regions (Johnson
et al., 2010; Zhou et al., 2020b). Mesoscale eddies can pump subsurface nutrient-rich water into the upper Ez and enhance
surface Chl *a* based on a long-term dataset of the Chl *a* anomaly corresponding to eddy properties (e.g., SLA, amplitude and
eddy rotation speed) in the oligotrophic SCS (He et al., 2016). Besides Chl *a*, POC concentrations and [234]Th deficits relative
to [238]U were also significantly enhanced in the upper 25 m by impacts from cyclonic eddies in the oligotrophic SCS where the
nutrient concentrations were observed to be quite low (Zhou et al., 2020b). This enhancement of biomass would be amplified
by the interplay of typhoons and cyclonic eddies (Liu et al., 2019). [15]N-isotopic results also indicate that subsurface nitrate is
an important external nutrient impacting export production (Yang et al., 2017). The nutrients from underlying waters may thus
play an important role in supporting POC export from the NDL.

As the potential availability of subsurface nutrients was determined by the depth of the nutricline and the nutrient
concentration in subsurface water (Moutin and Raimbault, 2002; Mouriño‑Carballido et al., 2021), we subsequently examined
relationships between partitioned POC export fluxes and the depth of the top of the nutricline, and subsurface DIN and DIP
concentrations below the Ez at 100, 125 and 150 m where biological uptake might be negligible (Fig 9d-j). The moderately
positive correlation ($R^2 = 0.4$, $p < 0.05$) between the depth of the top of the nutricline and POC export fluxes at the NDL base
(Fig 9d) suggests that shallower nutriclines might facilitate subsurface nutrient intrusion into the upper Ez, and subsequently
stimulate higher POC export fluxes in the upper nutrient-depleted ecosystems. Besides the nutricline, POC export fluxes at the
NDL base were also correlated ($R^2 \geq 0.4$) with DIN and DIP concentrations in the subsurface water near or below the Ez base
(Fig 9e-j). The positive relationship thus suggests that POC export fluxes in the upper nutrient-depleted Ez are also highly
associated with subsurface nutrient levels. Taken together, we confirm that subsurface nutrients significantly influence the
POC export flux at the NDL base.

#### 4.3.2 Nutrient sources diagnosed via [15]N-isotopic mass balance

As the timescale of [234]Th-[238]U disequilibrium was not instantaneous, any episodic intrusion events before sampling (~ 20 days)
could be recorded. Due to the limited Kuroshio intrusion into the SCS basin during the summer, the minor lateral transport of





nutrients by Kuroshio waters could not supply new N over the study area (Du et al., 2013). Thus, we assume that air-derived

nitrogen (i.e., NF and atmospheric nitrogen deposition [AND]) and upwelled nitrate are the major sources of new N supporting

the high POC export from the NDL. Using a two-endmember mixing model based on the $^{15}$N-isotopic balance (Kao et al.,

2012; Böttjer et al., 2017), we can evaluate the relative contribution of these two plausible sources of new N to support the

particle export at sites SEATS and SS1 using the following equations:

$$F_{PN} = F_{NO_3^-} + F_{Air} \qquad (10)$$

$$\delta^{15}N_{PN} \times F_{PN} = \delta^{15}NO_3^- \times F_{NO_3^-} + \delta^{15}N_{Air} \times F_{Air} \qquad (11)$$

where $F_{PN}$, $F_{NO_3^-}$ and $F_{Air}$ represent, the fluxes of total PN, PN contributed by upwelled DIN from the subsurface and nitrogen

from the atmosphere (i.e., N$_2$ fixation and AND), respectively, and $\delta^{15}NO_3^-$ and $\delta^{15}N_{Air}$ denote the endmembers of $\delta^{15}N$ for DIN

in subsurface waters and air-derived N, respectively.

$\delta^{15}N_{Air}$ was chosen as -1.1‰ by considering the influences of both NF and AND following Yang et al. (2022). The $\delta^{15}NO_3^-$

values in the upper water column of the SCS basin show little spatial and temporal variability, and average 4.7±0.4‰ at 100

m (Yang et al., 2017; Yang et al., 2022).

$F_{NO_3^-}$ was estimated to be about 59-67% of the total flux at the NDL base, and 86-98% at the Ez base at station SEATS. The

proportion was higher (84-96%) at 50 m within the NDL and nearly 100% at 100 m close to the Ez base at station SS1. The

differences in isotopic compositions of $\delta^{15}N_{PN}$ in the NDL should be a function of the relative contributions of nutrient sources.

Little variability in the regional NF rate suggests that differences in NF would not lead to such a discrete pattern of $\delta^{15}N_{PN}$

compositions near the NDL base between sites, except when influenced by Kuroshio waters (Lu et al., 2018). However, Gao

et al. (2020) clarified the spatial variability in AND in the SCS basin showing the aerosol NO$_3^-$ concentration at station SEATS

is nearly twice that at station SS1 which lies relatively far away from the continent. In addition, typhoon Mun (Fig 1) and three

anti-cyclonic eddies (Fig S4) influenced the water surrounding station SS1 before our visit in this region. In this regard, the

relatively elevated contribution of subsurface DIN at station SS1 might be attributed to the decrease in AND and event-driven

subsurface DIN intrusion. Despite the variability of $\delta^{15}N_{PN}$ between stations, our results suggested a major contribution of

subsurface DIN in the SCS basin based on the isotopic balance. These estimates indicate that POC export fluxes supported by

subsurface DIN are comparable, and even more important than those supported by NF and AND at the base of NDL where the

DIN concentration is usually below detection. The differences in $\delta^{15}N_{PN}$ at both stations SEATS and SS1 gradually disappeared

with increasing depth because the new nutrients were predominantly sourced from the nutrient-rich subsurface waters near the

base of the Ez. This enhanced contribution of subsurface nutriens is consistent with results from prior studies (Kao et al., 2012;

Yang et al., 2017) that indicate subsurface nutrients contribute to more than 90% of the export production at the Ez base.





To validate our estimates based on the [15]N-isotopic balance, we also compared the reported fluxes of NF and AND in the SCS basin to the measured PN fluxes from the sediment trap at 50 m (about 2.8 mmol C m[-2] d[-1] and 0.42 mmol N m[-2] d[-1], assuming a C/N ratio of 6.6 in sinking particles) at station SEATS in this study. The average NF rate was 0.06 mmol N m[-2] d[-1] (Kao et al., 2012; Chen et al., 2014) and the AND was 0.14 mmol N m[-2] d[-1] (Yang et al., 2014; Kim et al., 2014). The contribution of NF and AND to the measured PN flux at 50 m is estimated to be 48%. This mass-based estimate is consistent with the results derived from the isotopic balance.

In summary, we conclude that, compared to external N inputs from the atmosphere, nutrient intrusion from the subsurface is one of the major contributors supporting POC export fluxes at the NDL base in the oligotrophic SCS basin, and NF and AND may also contribute substantially to POC export flux at the NDL base.

We thus speculate that the episodic event-driven nutrient upwelling from the subsurface to the surface nutrient-depleted ecosystem stimulates the growth of planktonic organisms and elevates the particle scavenging rate in the oligotrophic SCS, which could be reflected in the [234]Th whose activities integrate the impacts of processes occurring over several months.

## 5 Conclusions

With the aid of high depth resolution [234]Th sampling, [234]Th and POC fluxes at both the NDL and Ez bases were estimated in the oligotrophic SCS basin during the summer of 2017. Although DIN was exhausted in the NDL, [234]Th-based POC export fluxes at the NDL base were estimated to be 1.1±0.5-4.3±1.2 mmol C m[-2] d[-1], which is comparable to those at the Ez base (1.2±0.5-3.0±0.8 mmol C m[-2] d[-1]). The relationship between POC export flux and nutrients was diagnosed: spatially, the POC export flux at the Ez was elevated at stations with shallow nutriclines, corresponding to high nutrient inventories (1.7±0.4-3.0±0.8 mmol C m[-2] d[-1]) relative to stations with low nutrient inventories (1.2±0.5-2.2±1.2 mmol C m[-2] d[-1]). More than 50% of the relatively high particle export occurring at the NDL base was verified by N-isotopes to be supported by DIN from the subsurface. It thus indicated that other pathways (e.g., episodic events) might be important for nutrient intrusion into the Ez. The higher POC export flux resulted from shallow-nutricline derived higher nutrient stocks and biomass in the Ez. We thus hypothesize that subsurface nutrients might act as the primary regulator of POC export fluxes at both the Ez and NDL bases on a seasonal timescale. The reduced export flux under the background of higher surface temperature and stronger stratification further implies that sea surface warming might lower the efficiency of the biological pump.

**Data Availability.**

All data accessed from *in situ* observations (i.e., temperature, salinity, fluorescence-based Chl *a*, [234]Th, POC and nutrients) are currently for review and will be available at National Science Data Bank (https://www.scidb.cn/en) with DOI. DOI number will be provided before the acceptance of this manuscript. The speeds of horizontal water current from May to August, 2017 and 2019 were obtained from the Copernicus Marine Environment Monitoring Service (CMEMS,



https://marine.copernicus.eu/). The vertical speeds of water current and diffusive (*Kz*) was derived from China Sea Muti-Scale Ocean Modeling System (CMOMS, https://odmp.ust.hk/cmoms/).

**Supplement.**

Additional figures referenced in text: **Figure S1**. Relationship between bottle-derived Chl a (Y-axis) and CTD fluorescence-based Chl a (X-axis). **Figure S2**. Surface distribution of monthly sea level anomalies (SLA, a) and eddy kinetic energy (EKE, b) with water currents during the cruise determined from modeling work. The SLA and EKE indicated stations SEATS, A1 and C1 experienced impacts of the mesoscale eddies. **Figure S3**. Climatological sea surface temperature anomalies in the SCS

during June from the China Sea Multi-scale Ocean Modeling System (CMOMS). Stations C1 and A2, impacted by cold water sourced from the southwest SCS basin during the survey, are shown. **Figure S4**. Surface distributions of monthly sea level anomalies (SLA) during the summer of 2019 with water currents from modeling work. The SLA show that station SS1 was impacted by mesoscale processes for at least one week before our visit (July 13th, 2019).

**Competing interests.**

The authors declare that they have no conflict of interest.

**Author contribution.**

All authors have been involved in the writing of the paper and have approved the final submitted manuscript. Yifan Ma and Minhan Dai are major contributors to the study's conception, data analysis and drafting the paper. Kuanbo Zhou, Weifang

Chen and Junhui Chen contributed significantly to cruise design, sample collections and/or data acquisition. Jin-Yu Terence Yang contributed substantially to isotopic data acquisition and analysis.

**Acknowledgements.**

This study was funded by the National Natural Science Foundation of China through grants No. 41890800 and 42188102, and by the National Basic Research Program of China (973 Program) through grant No.2015CB954000. Yifan Ma was supported

by a PhD Fellowship from the State Key Laboratory of Marine Environmental Science, Xiamen University. We thank Drs. Xianghui Guo, Peng Cheng and Yuyuan Xie who led the cruise as chief scientists, and Bangqing Huang with his group assisted Chloropyll-a data analysis. Zhongwei Yuan, Lifang Wang and Tao Huang are thanked for nutrient sampling and analysis. Silin Ni and Liguo Guo are thanked for helping with the collection of particulate samples. Qing Li and Li Tian are also thanked for




beta and POC/PN analyses, respectively. Yangyang Zhao, Zhongwei Yuan and Chuanjun Du are thanked for their valuable
comments. We are grateful to the crew of the R/V Tan Kah Kee along with its staff for their help during the cruise.

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
