# Peer review of "Partitioning of carbon export in the euphotic zone of the oligotrophic South China Sea"

_Biogeosciences, 2022_

## Author Comment (AC4)

Yifan Ma[1], Kuanbo Zhou[1]. Weifang Chen[1], Junhui Chen[1], Jin-Yu Terence Yang[1] & Minhan Dai[1]

[1]State Key Laboratory of Marine Environmental Science, College of Ocean and Earth Sciences, Xiamen University, Xiamen, 361102, China

*Correspondence to*: Minhan Dai (mdai@xmu.edu.cn)

**Anonymous Referee #2**

The article submitted by Ma et al investigates carbon export from the euphotic layer of the South China Sea, considering two layers (nutricline and euphotic layers). Carbon exports were calculated based on $^{234}$Th particulate fluxes and POC:$^{234}$Th ratio. The authors made a complicated discussion on the potential origin of nitrogen sources based on $^{15}$N-isotopic budget. The article is rather difficult to follow as the description of the dataset is not clear, and the sections are not always in the appropriate order. For example, it is really strange to discuss the impacts on physical transport on $^{234}$Th fluxes, whereas there were several pages where the $^{234}$Th fluxes and derived product were extensively discussed.

**[Response]:** We appreciate the constructive comments from the reviewer. Our manuscript is being thoroughly revised according to the reviewer's comments in order to optimize the discussion and the logic flow, and to enhance the readability. To do so, we have made a new table (Table R1) for better describing the dataset being used as suggested by the reviewer. In addition, the discussion of the physical transport on $^{234}$Th flux (section 4.1) will be moved to the methods part, before the $^{234}$Th flux and derived fluxes are discussed.

Table R1: Sampling logs and site information along with the accessed parameters and their utilizations.

| Station | Arriving time | Latitude [°N] | Longitude [°E] | Bottom depth [m] | Parameters | | Data utilizations | |
|---|---|---|---|---|---|---|---|---|
| | | | | | Total $^{234}$Th | Trap | Partitioning POC flux estimate | Nutrient source diagnosis |
| SEATS | 2017-06-07 00:06 | 18 | 116 | 3907 | √ | √ | √ | √ |
| A1[*] | 2017-06-11 23:55 | 16 | 116 | 4205 | √ | | √ | |
| SS1 | 2017-06-12 20:08 | 14 | 116 | 4107 | √ | | √ | |
| H06 | 2017-06-20 02:28 | 14.1 | 116 | 4289 | √ | | √ | |
| H08 | 2017-06-20 07:51 | 13.9 | 116 | 4063 | √ | | √ | |
| H01 | 2017-06-20 23:41 | 14 | 116.1 | 4139 | √ | | √ | |
| H11 | 2017-06-21 05:18 | 14 | 115.9 | 4297 | √ | | √ | |
| B1 | 2017-06-22 11:43 | 14 | 113 | 2537 | √ | | √ | |
| C1 | 2017-06-23 04:40 | 12 | 113 | 4313 | √ | | √ | |
| A2 | 2017-06-24 03:05 | 12 | 116 | 4079 | √ | | √ | |
| B2 | 2017-06-24 21:42 | 14 | 117 | 3947 | √ | | √ | |

[*] Sampling station might be influenced by the typhoon event passing through the South China Sea. Station A1 was visited after typhoon Merbok, which was generated on June 9, 2017 at 13.1°N, 119.8°E in the southern China Sea. Merbok landed on June 12 at 27.5°N, 117.3°E.

30

The dataset: there is a need of a table that presents clearly sampling, which station / when / what was measured (water column, trap). It is indicated that the cruise took place from June, 5 to 27, 2017. But typhoon Merbok occurred the 10th. "before our field campaign". This needs to be clarified. In case of a typhoon had occurred during sampling, one could expect it had impacted the water column and chemical budget. In addition, how could it be possible to use the described $^{234}$Th model which

35 is a steady-state model

**[Response]:** Thanks for the advices from the reviewer, and a new table of the sampling information will be included in the MS and is shown above.

Note that we did not conduct samplings before or during the typhoon, thus it is impossible for us to build up a non-steady state (NSS) model for $^{234}$Th flux estimation. However, we reasoned that a SS model is in order in the condition under study as

40 the Chl $a$ concentration was not significantly enhanced under the impact of typhoon as shown by the remote sensing derived 8-day averaged surface Chl $a$ (Figure R1). Indeed, the SS $^{234}$Th fluxes remained pretty low, mostly <800 dpm m$^{-2}$ d$^{-1}$ during our study, suggesting that again, export was not much beyond steady state as shown in many prior studies (e.g., Resplandy et al., 2012; Savoye et al., 2006). Additional justification of the SS model will be added during the revision.

[Figure]

Figure R1: Satellite-derived the 8-day averaged surface Chl $a$ in the SCS basin during June 2017, showing that sea surface Chl $a$ concentration was little enhanced during our ship-based sampling period. Note that Station A1 was visited after typhoon Merbok, which was generated on June 9, 2017 at 13.1°N, 119.8°E in the southern China Sea. Merbok landed on June 12 at 27.5°N, 117.3°E.

Four stations (H01, H06, H08, H11) were sampled around SS1 to check the spatial variability of $^{234}$Th. But it is indicated later that the mega station SS1 was revisited during August 2019 (after a second typhoon Mun, July, 1th) with trap deployment. A clarification must then to be made on what was sampled / when / where.

[Response]: We apologize that we were not clear enough in describing the multiple events happened prior to and post our sampling campaign. We have now included such information in the Table R1 with clarifications throughout the revised MS.

55 Note that our ship-based sampling occurred from June 5$^{th}$ to June 27$^{th}$, 2017 with samplings at station SS1 and its surrounding stations (H01, H06, H08 and H11) on June 12$^{th}$, 2017 We did deploy sediment traps at Station SS1 but unfortunately, the traps were not retrieved. We thus used a trap results deployed two years later on July 13$^{th}$, 2019 From Station SS1. It must be pointed out that the data accessed from sediment traps deployed at Station SS1 in 2019 was only utilized to evaluate the contribution of subsurface nutrient by $\delta^{15}N_{PN}$.

60

High resolutions profiles: the authors made a great announcement about high resolution profiles. In fact, there are only two, t detailed profiles: SEAT and SS1. The other profiles have a less resolution, and, except for the lower total $^{234}$Th values at about 25 meters at station A1, the profiles of total $^{234}$Th are not so different. It would be interesting that the authors reduce the depth resolution of the SEATS and SS1 profiles to compare the estimated $^{234}$Th.

65

[Response]: The reviewer is right that the 10-m vertical interval samplings were only conducted at stations SEATS and SS1. We will clarify this in our revision. Following suggestions, we estimated $^{234}$Th fluxes at the Ez base by reducing the vertical resolution to a 25-m interval, being 490±60 and 655±71 dpm m$^{-2}$ d$^{-1}$ respectively for station SEATS and SS1 compared to 522±43 and 631±48 dpm m$^{-2}$ d$^{-1}$ under the high-resolution sampling mode. The low-resolution sampling thus might induce a

70 less than 6% of uncertainty for the $^{234}$Th flux. However, the high-resolution sampling is essential in order to examine the partitioning of carbon export in the upper water column, especially for the oligotrophic ocean characteristic of the low export flux. Based on high-resolution total $^{234}$Th pattern at stations SEATS and SS1, we first determined $^{234}$Th deficit in the NDL, showing the substantial particle scavenging and POC export at the NDL base at both stations, and we subsequently found similar pattern at the rest of stations and estimated the partitioning in POC export flux between two layers. The reduced

75 sampling resolution might introduce some additional uncertainty to estimates of $^{234}$Th flux and $^{234}$Th-derived POC export flux, but would not change our main conclusion that the base of NDL is the hotspot for particle scavenging and POC export. We will include the above clarification and reasoning in our revision.

Export model: from equation (2), the authors need to produce the two equations relative to the export estimate for the NDL-

80 and NRL-layers, respectively. Use directly the symbol for Fndl and Fnrl. There is no need to use layer i /i-1, that only complicate the model presentation. Also from Fig 2, it seems that calculations are done for each box, but from the text it is less clear that the fluxes from NRL-layer is calculated considering only the lower box or the whole water column above the euphotic layer limit. Figure 2 needs also to be improved: if total $^{234}$Th activities are related to U activities, what means 'absorb particles, total TH already includes particulate phase. The figure needs to be corrected.

85

[Response]: We agree with the reviewer's comments for the Figure 2. We actually calculated $^{234}$Th flux at the export horizons

of NDL base and the euphotic zone (Ez) bottom, with the integration carried out between 0-NDL base and 0-Ez bottom (not the lower box as mentioned by the reviewer). Here we use symbol $F_{NDL}$ and $F_{Ez}$ as suggested by the reviewer. In order to make the statement clearer, we will revise the main text to emphasize that the flux at the Ez bottom is integrated from the whole box

90    from surface to the Ez bottom.

The reviewer is also right that we only measured total $^{234}$Th activities during the cruises, and we will delete the "particles" in the figure and change "$A^{dissolved}$" into "$A^{total}$" as suggested by the reviewer (see the details in Figure R2).

[Figure]

Figure R2: Schematic of the $^{234}$Th model under the two-layer nutrient structure. All terms are defined in Equations (2)-(6).

95

The conversion of $^{234}$Th particulate fluxes in POC fluxes: the conversion is done using the POC/$^{234}$Th. The recommendation is to use the large particle ratio. In this work, the authors use the ratio obtained from bottle waters, that correspond to fine particles. The authors need to better argument the choice. The comparison with the trap ratio seems to be biased as trap was

100    done in summer, no during the same sampling cruise. The authors need to be clearer on this aspect. If confirmed, it means that some paragraphs are not justified.

**[Response]:** Bottle filtration and trap deployment for POC/$^{234}$Th were done at Station SEATS during the same cruise (See Table R1). Bottle-derived POC/$^{234}$Th ratios at the depth of 50 m and 100 m were respectively 4.4±0.6 and 3.8±0.3 μmol C

105    dpm$^{-1}$ compared to 4.4±0.6 and 3.2±0.4 μmol C dpm$^{-1}$ from trap samples. We thus confirmed that bottle-derived POC/$^{234}$Th was comparable with those derived from trap samples during this cruise. This is consistent with what Zhou et al. (2020) found showing that POC export fluxes based on bottle POC/$^{234}$Th was comparable with trap POC fluxes measured before. More importantly, it was impossible to deploy sediment traps at all stations due to practical reasons. For consistency with prior studies in the region (e.g., Cai et al., 2008; Zhou et al., 2013; Cai et al., 2015; Zhou et al., 2020), we primarily used bottle

110    derived POC/$^{234}$Th in estimating POC export fluxes as we routinely did in our prior work.

Th/POC flux estimates: most of the article is based on fluxes, but the authors treated data as it was rather instantaneous fluxes,

which is clearly not the case. Considering the half-life of $^{234}$Th, a deficit of $^{234}$Th in the water column represent a flux story of several weeks. The only way to have more "instantaneous" fluxes is to repeat profiles at the same station which was not done here. Therefore, it is the main problem with the article. The authors discussed a lot fluxes and potential nutrient sources, but the errors on the fluxes estimate do not support the discussion. There is an over-interpretation of the dataset and the derived fluxes to support the hypothesis of the authors.

[Response]: We completely agree with the reviewer that $^{234}$Th-derived POC export flux is not instantaneous but with a timescale of weeks to months, and understand the reviewer's concern on the potential issues associated with the correlation between "instantaneous" nutrient and time integrated POC flux. In order to match the time scale between nutrient and POC fluxes, we also correlated $^{234}$Th-derived POC flux with the model-derived monthly average of nutrients during summer (Figure R3, Du et al., 2021). The correlation in between is indeed statistically significant (P<0.05). This suggests that under the oligotrophic condition of the present study, the euphotic layer characterized by low biological productivity, and the system under study is pretty much under steady state. The overall low $^{234}$Th flux as we explained in our above responses to the reviewer, also supports this notion. In addition, we examined the $\delta^{15}N_{PN}$ value measured in several previous studies in the region (e.g., Kao et al., 2012; Yang et al., 2017; Yang et al., 2022). Taken together, we contend that the conclusion of the subsurface nutrient supported largely is a well plausible interpretation of the dataset. Having said, we will fully consider the comments from the reviewer and revise our MS accordingly.

[Figure]

Figure R3: Relationship between POC export fluxes at the NDL base (black dots) and Ez base (grey dots) vs. the model-derived depth of the top of the nutricline (top) and DIN concentration in the subsurface water at 100 m (bottom).

Others comments: most figures need to be improved and some data combined differently. What is the interest of figure 3?

**[Response]:** Thanks for the comments, we have revised the figures based on the suggestion above from the reviewer. As our figure 4 has shown the vertical profiles of T and S, here we deleted the figure 3 to simplify the discussion.

**References**

Cai, P. H., M. H. Dai, W. F. Chen, T. T. Tang, and K. B. Zhou: On the importance of the decay of $^{234}$Th in determining size-fractionated C/$^{234}$Th ratio on marine particles, *Geophys Res Lett*, 33, 23, 10.1029/2006gl027792, 2006.

Cai, P. H., D. C. Zhao, L. Wang, B. Q. Huang, and M. H. Dai: Role of particle stock and phytoplankton community structure in regulating particulate organic carbon export in a large marginal sea, *J Geophys Res-Oceans*, 120, 2063-2095, 10.1002/2014jc010432, 2015.

Cai, P. H., W. F. Chen, M. H. Dai, Z. W. Wan, D. X. Wang, Q. Li, T. T. Tang, and D. W. Lv: A high-resolution study of particle export in the southern South China Sea based on $^{234}$Th : $^{238}$U disequilibrium, *J Geophys Res-Oceans*, 113, C04019, 10.1029/2007jc004268, 2008.

Du, C. J., R. Y. He, Z. Y. Liu, T. Huang, L. F. Wang, Z. W. Yuan, Y. P. Xu, Z. Wang, and M. H. Dai: Climatology of nutrient distributions in the South China Sea based on a large data set derived from a new algorithm, *Prog Oceanogr*, 195, 102586, 10.1016/j.pocean.2021.102586, 2021.

Kao, S. J., Terence Yang, J. Y., Liu, K. K., Dai, M., Chou, W. C., Lin, H. L., Ren, H.: Isotope constraints on particulate nitrogen source and dynamics in the upper water column of the oligotrophic South China Sea. *Global Biogeochem Cycles*, 26: GB2033, 10.1029/2011GB004091, 2012.

Resplandy, L., Martin, A. P., Le Moigne, F., Martin, P., Aquilina, A., Mémery, L., Lévy, M. and Sanders, R.: How does dynamical spatial variability impact $^{234}$Th-derived estimates of organic export? Deep Sea Research Part I, 68: 24-45, doi: 10.1016/j.dsr.2012.05.015, 2012.

Savoye, N., C. Benitez-Nelson, A. B. Burd, J. K. Cochran, M. Charette, K. O. Buesseler, G. A. Jackson, M. Roy-Barman, S. Schmidt, and M. Elskens: $^{234}$Th sorption and export models in the water column: A review, *Mar Chem*, 100, 234-249, 10.1016/j.marchem.2005.10.014, 2006.

Yang, J. Y. T., Kao, S. J., Dai, M., Yan, X., Lin, H. L.: Examining N cycling in the northern South China Sea from N isotopic signals in nitrate and particulate phases. *J Geophys Res-Biogeoscience*, 122: 2118-2136, 10.1002/2016JG003618, 2017.

Yang, J. Y. T., Tang, J. M., Kang, S., Dai, M., Kao, S. J., Yan, X., Xu, M. N., Du, C.: Comparison of nitrate isotopes between the South China Sea and western North Pacific Ocean: Insights into biogeochemical signals and water exchange. *J Geophys Res-Oceans*, 127: e2021JC018304, 10.1029/2021JC018304, 2022.

Zhou, K. B., M. H. Dai, S. J. Kao, L. Wang, P. Xiu, F. Chai, J. W. Tian, and Y. Liu: Apparent enhancement of $^{234}$Th-based particle export associated with anticyclonic eddies, *Earth Planet Sc Lett*, 381, 198-209, 10.1016/j.epsl.2013.07.039, 2013.

Zhou, K. B., M. H. Dai, K. Maiti, W. F. Chen, J. H. Chen, Q. Q. Hong, Y. F. Ma, P. Xiu, L. Wang, and Y. Y. Xie: Impact of physical and biogeochemical forcing on particle export in the South China Sea, *Prog Oceanogr*, 187, 102403, 10.1016/j.pocean.2020.102403, 2020.

170

---

## Author Response (AR1)

*Re: "Partitioning of carbon export in the upper water column of the oligotrophic South China Sea" by Ma et al.*

*13th March, 2023*

5  Dear Editor,

Enclosed please find our revised manuscript "Partitioning of carbon export in the upper water column of the oligotrophic South China Sea" by Ma et al.

10  We have carefully considered the comments and suggestions from both reviewers, and have thoroughly revised our MS accordingly. To highlight some of these revisions, we have moved the discussion of the influence of physical transport on the $^{234}$Th flux calculation to the methods section. We have also added a table to clarify the sampling strategies as suggested by both reviewers. We have additionally expanded our discussion on the relationship between $^{234}$Th-derived POC flux and nutrients in the water column. 15  Lastly, we have improved the quality of the figures for better illustrations. More detailed revisions are explained in the enclosure.

We would like to take this opportunity to thank you for handling the paper. We would also like to thank the reviewers for their constructive comments and suggestions, which significantly improved the quality 20  of the paper. We sincerely hope that our revision will meet the high standard of Biogeosciences.

Sincerely,

Minhan Dai

Corresponding author

25    State Key Lab of Marine Environmental Science

Xiamen University

Xiamen 361005, China

**Anonymous Referee #1**

30 Ma et al. calculated POC export fluxes at the base of the NDL and Ez, as well as discussed the NDL's nutrient source. The data is treasurable for understanding nutrient dynamics and the carbon cycle. The outcome is reliable, and the manuscript is well-organized. However, some points must be clarified before accepting for publication. There are also a number of typos. My specific recommendations are listed below.

**[Response]:** We appreciate the positive comments from the reviewer. Our point-by-point responses are listed below.

**Specific recommendations**

35 My biggest concern is about the method calculating the physical transport flux. In eq. 8, V is part of the tendency term shown in eq. 3. To calculate the horizontal transport flux in the NDL or Ez, it needs to implement an integration over the depth. Whereas, the vertical flux is calculated as the wC, where w is the vertical velocity and C is the concentration of the tracer. It isn't necessary to calculate the "integrated vertical transport flux" over the NDL or Ez as shown in L306. Please recheck your method. I listed some references that introduce the method to calculate transport fluxes. The authors need to introduce how

40 they calculated the horizontal and vertical fluxes clearly.

**[Response]:** The reviewer is correct that integration for the calculation of vertical transport flux of $^{234}$Th is unnecessary. We have double checked the calculation of the vertical transport flux of $^{234}$Th at the base of the NDL and the Ez. The $w$ and $K_z$ at the base of Ez (110 m) were -0.10 m d$^{-1}$ and 0.86 m$^2$ d$^{-1}$, respectively from Gan et al. (2016). The $^{234}$Th activity at 125 m and 100 m was 2.44±0.04 and 2.50±0.02 dpm L$^{-1}$, respectively at Station SS1. The physical term $V$ of vertical $^{234}$Th flux was

45 estimated to be -2.0±0.4 dpm m$^{-2}$ d$^{-1}$ at the base of Ez based on the following equation (adapted from McGillicuddy et al., (2003) as recommended by the reviewer):

$$-V = \left[ w \times (A_{^{234}Th@100m} - A_{^{234}Th@125m})_{Z_{Ez}} \right] + \left[ K_z \times \frac{(A_{^{234}Th@125m} - 2 \times A_{^{234}Th@Z_{Ez}} + A_{^{234}Th@100})}{\Delta z^2} \right]$$

where, $\Delta z$ is the distance between sampling depths. The estimated vertical flux of $^{234}$Th at the NDL base was -11.4±0.1 dpm

50 m$^{-2}$ d$^{-1}$ at station SS1. Therefore, the physical term could still be neglected. We will revise the text as: "The vertical transport fluxes were estimated to be -2.0±0.4 and -11.4±0.1 dpm m$^{-2}$ d$^{-1}$ at the base of Ez and NDL, respectively, accounting for <10% of the vertical scavenging fluxes at corresponding layers at the station SS1, which can be considered to be negligible."

In section 4.3.2, the authors calculated the mass balance of $^{15}$N (Eqs. 10, 11). In my understanding, PN which denote particulate nitrogen should be interpreted when it occurred for the first time. It is not clear how to calculate the 3 unknowns (Fpn, Fno3, Fair) in two equations. Please introduce the calculation carefully.

**[Response]:** Following suggestions, we will explain "PN" at its first appearance, which will read: "POC and particulate nitrogen (PN) concentrations were determined by an Elemental Analyzer-Isotope Ratio Mass Spectrometer (EA-IRMS) system…"

In addition, we have rephrased the parameters and changed Equations 10 &11 as follows: "

$$1 = f_{NO_3^-} + f_{Air} \tag{10}$$

$$\delta^{15}N_{PN} = \delta^{15}N_{NO_3^-} \times f_{NO_3^-} + \delta^{15}N_{air} \times f_{air} \tag{11}$$

where, $f_{NO_3^-}$ and $f_{Air}$ represent the fraction of PN export contributed by upwelled DIN from the subsurface and by air-derived nitrogen from nitrogen fixation and atmospheric nitrogen deposition. $\delta^{15}N_{NO_3^-}$ and $\delta^{15}N_{air}$ denote the endmembers of $\delta^{15}N$ for DIN in subsurface waters and air-derived N, respectively."

The authors discovered that horizontal transport flux accounts for 20% of total flux. However, the fraction is not negligible. Some stations were shown to be influenced by eddy activities. It is worthwhile to consider the horizontal transport of eddies whose effect is not only vertical. There are some studies discussed the horizontal transport of particles in eddies e.g. Wang et al., 2018 http://dx.doi.org/10.1029/2017JC013623, Ma et al., 2021, http://dx.doi.org/10.1016/j.pocean.2021.102566. Can you separate the nutrients trapped in the cyclonic eddy and transported with the eddy (horizontal transport) and local uplifted nutrients (vertical transport)? Stations B1 and C2 may be affected by the upwelling off the coast of Vietnam.

**[Response]:** We appreciate these important comments from the reviewer aiming for improving the flux estimate. Meanwhile, we have to recognize that the horizontal transport flux of 20% was an upper limit of estimates, which is overall comparable with the magnitude of the uncertainty from $^{234}$Th measurements (could be >10%). Therefore, the horizontal flux of <20% of the total flux has been typically omitted in many prior studies given the difficulty in the estimation therein (e.g., Buesseler et al., 2020; Wei et al., 2011). We will add such reasoning in our revision.

We agree with the reviewer that mesoscale eddies impact flux estimations. Unfortunately, such effects of eddies cannot be resolved from the present study. We will add in our revision such potential impacts of mesoscale eddies. The reviewer also made significant comments on different pathways of nutrient trapping. But again, distinguishing these processes is extremely challenging (Guo et al., 2017; Zhao et al., 2021). Nevertheless, we considered the comments from the reviewer and will add the following text: "It is also worthwhile considering the influences from mesoscale and sub-mesoscale processes in the SCS basin. Prior studies showed the concurrence of the vertical transport of particles supported by locally uplifted nutrients and the

horizontal transport of particles supported by the nutrients trapped in eddies (Wang et al., 2018, Ma et al., 2021). In this study, we found enhanced POC export fluxes at stations with high nutrient inventories and inferred that the POC export flux might also be supported by nutrients from the subsurface based on the signal of $\delta^{15}N_{PN}$. However, our current study was unable to diagnose the pathways of nutrients fuelling the primary and export production, which needs further studies."

**Minor concerns:**

L36: Siegel et al., 2021

**[Response]:** Corrected.

L41-42: Need references

**[Response]:** Accepted. We will add the relevant references in the revision. "(Benitez-Nelson et al., 2001; Cai et al., 2015; Zhou et al., 2020)."

L53: the references are not recent ones. Don't use the word recently.

**[Response]:** Accepted and revision will be made accordingly.

100  Figure 1: Denote the shading and add a color bar.

**[Response]:** Accepted. We redraw the map and add a color bar in the revised version (see the details in Fig. R1).

[Figure]

**Figure R1:** Map of the South China Sea (SCS) with sampling stations during June 2017. Yellow diamonds denote mega
105  stations (SEATS and SS1) where high-resolution sampling was conducted at a 10-m interval in the euphotic zone; red circles
denote regular stations where samples were collected at typical sampling depths of 5, 25, 50, 75 and 100 m. The general
circulation pattern (adapted from Liu et al., 2016) is also shown. The dominant summer currents are denoted by black dashed
arrows. The dark blue dashed line denotes the path of typhoon Merbok (generated at the southeastern part of the SCS on June
9th, 2017).

110 Please consider to make a new table to show the location, water depth, sampling depth, sampling time etc.

**[Response]:** As suggested by the reviewer, two tables: the location of sampling stations with arriving and leaving time, water bottom depth, parameters and data utilization in the Table R1 and sampling depth with the [234]Th and POC data in Table R2 are available in the revision.

115 Table R1: Sampling logs and site information along with the accessed parameters and their utilizations.

| Station | Arriving time | Latitude [ºN] | Longitude [ºE] | Bottom depth [m] | Parameters | | Data utilizations | |
|---|---|---|---|---|---|---|---|---|
| | | | | | Total [234]Th | Trap | Partitioning POC flux estimate | Nutrient source diagnosis |
| SEATS | 2017-06-07 00:06 | 18 | 116 | 3907 | ✓ | ✓ | ✓ | ✓ |
| A1* | 2017-06-11 23:55 | 16 | 116 | 4205 | ✓ | | ✓ | |
| SS1 | 2017-06-12 20:08 | 14 | 116 | 4107 | ✓ | | ✓ | |
| H06 | 2017-06-20 02:28 | 14.1 | 116 | 4289 | ✓ | | ✓ | |
| H08 | 2017-06-20 07:51 | 13.9 | 116 | 4063 | ✓ | | ✓ | |
| H01 | 2017-06-20 23:41 | 14 | 116.1 | 4139 | ✓ | | ✓ | |
| H11 | 2017-06-21 05:18 | 14 | 115.9 | 4297 | ✓ | | ✓ | |
| B1 | 2017-06-22 11:43 | 14 | 113 | 2537 | ✓ | | ✓ | |
| C1 | 2017-06-23 04:40 | 12 | 113 | 4313 | ✓ | | ✓ | |
| A2 | 2017-06-24 03:05 | 12 | 116 | 4079 | ✓ | | ✓ | |
| B2 | 2017-06-24 21:42 | 14 | 117 | 3947 | ✓ | | ✓ | |

*Sampling station might be influenced by the typhoon event passing through the South China Sea. Station A1 was visited after typhoon Merbok, which was generated on June 9, 2017 at 13.1ºN, 119.8ºE in the southern South China Sea. Merbok landed on June 12 at 27.5ºN, 117.3ºE.

120  Table R2: The list of total and particulate $^{234}$Th activity and POC concentration at sampling depth at stations

| Station | Latitude degree (N) | Longitude degree (E) | Depth m | Tot. $^{234}$Th dpm L$^{-1}$ | Tot. $^{234}$Th error dpm L$^{-1}$ | POC µmol L$^{-1}$ | Part. $^{234}$Th dpm L$^{-1}$ | Part. $^{234}$Th error dpm L$^{-1}$ |
|---|---|---|---|---|---|---|---|---|
| SEATS | 18 | 116 | 130 | 2.55 | 0.06 | 0.6 | 0.13 | 0.01 |
| SEATS | 18 | 116 | 100 | 2.47 | 0.06 | 0.8 | 0.20 | 0.01 |
| SEATS | 18 | 116 | 95 | 2.73 | 0.07 | 2.0 | 0.32 | 0.01 |
| SEATS | 18 | 116 | 85 | 2.41 | 0.05 | 2.1 | 0.39 | 0.01 |
| SEATS | 18 | 116 | 75 | 2.29 | 0.06 | 2.4 | 0.47 | 0.01 |
| SEATS | 18 | 116 | 65 | 2.30 | 0.06 | 2.2 | 0.43 | 0.01 |
| SEATS | 18 | 116 | 55 | 2.03 | 0.05 | 1.8 | 0.41 | 0.01 |
| SEATS | 18 | 116 | 45 | 2.22 | 0.06 | 1.3 | 0.30 | 0.01 |
| SEATS | 18 | 116 | 35 | 2.13 | 0.05 | 1.4 | 0.16 | 0.01 |
| SEATS | 18 | 116 | 25 | 2.30 | 0.05 | 1.6 | 0.12 | 0.01 |
| SEATS | 18 | 116 | 15 | 2.03 | 0.05 | 1.2 | 0.15 | 0.01 |
| SEATS | 18 | 116 | 5 | 2.27 | 0.05 | 1.1 | 0.11 | 0.01 |
| A1 | 16 | 116 | 100 | 2.59 | 0.05 | 1.1 | 0.26 | 0.01 |
| A1 | 16 | 116 | 75 | 2.47 | 0.05 | 2.5 | 0.26 | 0.01 |
| A1 | 16 | 116 | 50 | 2.17 | 0.05 | 1.8 | 0.29 | 0.01 |
| A1 | 16 | 116 | 25 | 1.70 | 0.25 | 1.3 | 0.15 | 0.01 |
| A1 | 16 | 116 | 5 | 2.34 | 0.06 | 1.3 | 0.11 | 0.01 |
| SS1 | 14 | 116 | 125 | 2.44 | 0.05 | 0.8 | 0.25 | 0.01 |
| SS1 | 14 | 116 | 110 | 2.42 | 0.10 | 0.9 | 0.27 | 0.01 |
| SS1 | 14 | 116 | 100 | 2.39 | 0.06 | 1.3 | 0.42 | 0.01 |
| SS1 | 14 | 116 | 95 | 2.50 | 0.06 | 1.3 | 0.41 | 0.01 |
| SS1 | 14 | 116 | 85 | 2.32 | 0.06 | 1.7 | 0.41 | 0.01 |
| SS1 | 14 | 116 | 75 | 1.98 | 0.06 | 1.3 | 0.30 | 0.01 |
| SS1 | 14 | 116 | 65 | 2.06 | 0.05 | 1.5 | 0.35 | 0.01 |
| SS1 | 14 | 116 | 55 | 2.35 | 0.05 | 1.4 | 0.23 | 0.01 |
| SS1 | 14 | 116 | 45 | 2.15 | 0.06 | 0.6 | 0.20 | 0.01 |
| SS1 | 14 | 116 | 35 | 2.04 | 0.05 | 1.3 | 0.14 | 0.01 |
| SS1 | 14 | 116 | 25 | 2.15 | 0.05 | 1.1 | 0.18 | 0.01 |

| Station | Latitude degree (N) | Longitude degree (E) | Depth m | Tot. $^{234}$Th dpm L$^{-1}$ | Tot. $^{234}$Th error dpm L$^{-1}$ | POC μmol L$^{-1}$ | Part. $^{234}$Th dpm L$^{-1}$ | Part. $^{234}$Th error dpm L$^{-1}$ |
|---|---|---|---|---|---|---|---|---|
| SS1 | 14 | 116 | 15 | 1.99 | 0.05 | 1.2 | 0.17 | 0.01 |
| SS1 | 14 | 116 | 5 | 2.15 | 0.07 | 1.2 | 0.19 | 0.01 |
| H06 | 14.1 | 116 | 100 | 2.41 | 0.05 | 1.4 | 0.50 | 0.01 |
| H06 | 14.1 | 116 | 75 | 2.05 | 0.05 | 1.5 | 0.42 | 0.01 |
| H06 | 14.1 | 116 | 50 | 2.33 | 0.05 | 1.1 | 0.19 | 0.01 |
| H06 | 14.1 | 116 | 25 | 2.21 | 0.05 | 1.0 | 0.13 | 0.01 |
| H06 | 14.1 | 116 | 5 | 2.27 | 0.05 | 1.0 | 0.11 | 0.01 |
| H08 | 13.9 | 116 | 100 | 2.39 | 0.05 | 1.4 | 0.30 | 0.01 |
| H08 | 13.9 | 116 | 75 | 2.15 | 0.05 | 1.9 | 0.30 | 0.01 |
| H08 | 13.9 | 116 | 50 | 2.25 | 0.05 | 1.4 | 0.23 | 0.01 |
| H08 | 13.9 | 116 | 25 | 2.21 | 0.05 | 1.1 | 0.25 | 0.01 |
| H08 | 13.9 | 116 | 5 | 2.27 | 0.05 | 0.9 | 0.16 | 0.01 |
| H01 | 14 | 116.1 | 100 | 2.45 | 0.05 | 1.8 | 0.53 | 0.01 |
| H01 | 14 | 116.1 | 75 | 2.25 | 0.05 | 1.3 | 0.22 | 0.01 |
| H01 | 14 | 116.1 | 50 | 2.29 | 0.05 | 1.8 | 0.34 | 0.01 |
| H01 | 14 | 116.1 | 25 | 2.25 | 0.05 | 1.6 | 0.24 | 0.01 |
| H01 | 14 | 116.1 | 5 | 2.10 | 0.05 | 1.3 | 0.15 | 0.01 |
| H11 | 14 | 116.1 | 100 | 2.46 | 0.05 | 1.3 | 0.30 | 0.01 |
| H11 | 14 | 116.1 | 75 | 2.23 | 0.04 | 1.1 | 0.40 | 0.01 |
| H11 | 14 | 116.1 | 50 | 2.25 | 0.05 | 1.3 | 0.13 | 0.01 |
| H11 | 14 | 116.1 | 25 | 2.29 | 0.05 | 1.0 | 0.08 | 0.01 |
| H11 | 14 | 116.1 | 5 | 2.09 | 0.04 | 1.0 | 0.11 | 0.01 |
| B1 | 14 | 113 | 100 | 2.44 | 0.05 | 1.4 | 0.23 | 0.01 |
| B1 | 14 | 113 | 88 | 2.08 | 0.04 | 2.0 | 0.52 | 0.01 |
| B1 | 14 | 113 | 75 | 2.30 | 0.08 | 1.8 | 0.41 | 0.01 |
| B1 | 14 | 113 | 50 | 2.21 | 0.05 | 1.4 | 0.40 | 0.01 |
| B1 | 14 | 113 | 25 | 2.24 | 0.04 | 1.2 | 0.08 | 0.01 |
| B1 | 14 | 113 | 5 | 2.24 | 0.05 | 0.9 | 0.06 | 0.01 |
| C1 | 12 | 113 | 100 | 2.55 | 0.05 | 0.7 | 0.21 | 0.01 |
| C1 | 12 | 113 | 88 | 2.49 | 0.05 | 0.8 | 0.25 | 0.01 |
| C1 | 12 | 113 | 75 | 2.38 | 0.04 | 1.9 | 0.30 | 0.01 |
| C1 | 12 | 113 | 50 | 2.05 | 0.04 | 1.5 | 0.23 | 0.01 |
| C1 | 12 | 113 | 25 | 2.10 | 0.09 | 1.6 | 0.25 | 0.01 |
| C1 | 12 | 113 | 5 | 2.06 | 0.04 | 2.1 | 0.29 | 0.01 |
| A2 | 12 | 116 | 100 | 2.63 | 0.05 | 1.1 | 0.21 | 0.01 |
| A2 | 12 | 116 | 88 | 2.21 | 0.04 | 0.9 | 0.42 | 0.01 |

| Station | Latitude degree (N) | Longitude degree (E) | Depth m | Tot. $^{234}$Th dpm L$^{-1}$ | Tot. $^{234}$Th error dpm L$^{-1}$ | POC μmol L$^{-1}$ | Part. $^{234}$Th dpm L$^{-1}$ | Part. $^{234}$Th error dpm L$^{-1}$ |
|---|---|---|---|---|---|---|---|---|
| A2 | 12 | 116 | 75 | 2.16 | 0.04 | 1.5 | 0.25 | 0.01 |
| A2 | 12 | 116 | 50 | 2.01 | 0.04 | 1.5 | 0.23 | 0.01 |
| A2 | 12 | 116 | 25 | 2.18 | 0.04 | 1.2 | 0.09 | 0.01 |
| A2 | 12 | 116 | 5 | 1.85 | 0.06 | 1.2 | 0.14 | 0.01 |
| B2 | 14 | 117 | 108 | 2.51 | 0.04 | 1.1 | 0.13 | 0.01 |
| B2 | 14 | 117 | 100 | 2.49 | 0.04 | 0.9 | 0.27 | 0.01 |
| B2 | 14 | 117 | 75 | 2.24 | 0.04 | 1.3 | 0.15 | 0.01 |
| B2 | 14 | 117 | 50 | 2.22 | 0.05 | 1.8 | 0.27 | 0.01 |
| B2 | 14 | 117 | 25 | 2.40 | 0.05 | 1.1 | 0.12 | 0.01 |
| B2 | 14 | 117 | 5 | 2.25 | 0.05 | 1.3 | 0.28 | 0.01 |

| Station | Latitude degree (N) | Longitude degree (E) | Depth m | Tot. $^{234}$Th dpm L$^{-1}$ | Tot. $^{234}$Th error dpm L$^{-1}$ | POC μmol L$^{-1}$ | Part. $^{234}$Th dpm L$^{-1}$ | Part. $^{234}$Th error dpm L$^{-1}$ |
|---|---|---|---|---|---|---|---|---|
| A2 | 12 | 116 | | | | | | |

Eq. 3 is the same as Eq. 1.

**[Response]:** Fixed. We will revise the equation as:

$$F_{Th}^{Ez} = \int_{0}^{Ez} \left( A_U - A_{Th} \right) \times \lambda dz \tag{3}$$

125

The font is too small in Figure 4.

**[Response]:** We appreciate the reviewer's comment. We have enlarged the font sizes as of below (see the details in Fig. R2).

[Figure]

130 **Figure R2:** Vertical profiles of temperature (a), salinity (b), dissolved inorganic nitrogen (nitrate + nitrite, DIN, c) and Chl *a* (d). The MLD (red dash), interpolated depth of DIN=0.1 μmol L$^{-1}$ (top of nutricline, yellow dash) and subsurface Chl *a* Maximum (SCM, green dash) are also shown.

135    Eq9. What's the delta x and delta y.

[Response]: We appreciate the reviewer's comment. Eq. 9 aimed to resolve the horizontally diffusive flux of $^{234}$Th. The $\Delta x$ and $\Delta y$ are the distance between the normal stations to evaluate the influences of physical terms ($\Delta y$ was the distance between station H06 and H08 and equal to 18 km) in this study.

We will explain the $\Delta x$ and $\Delta y$ in line 193 in our revision: "The $\Delta x$ and $\Delta y$ are the distance between the normal stations

140    to evaluate the influences of physical terms (i.e., $\Delta x$ is the distances between stations H01 and H11; $\Delta y$ is the distances between stations H06 and H08). $\Delta x$ and $\Delta y$ were equal to 18 km in this study."

**References**

Benitez-Nelson, C., K. O. Buesseler, D. M. Karl, and J. Andrews: A time-series study of particulate matter export in the North Pacific Subtropical Gyre based on $^{234}$Th: $^{238}$U disequilibrium, *Deep-Sea Res I*, 48, 2595-2611, 10.1016/S0967-0637(01)00032-2, 2001.

Buesseler, K. O., C. R. Benitez-Nelson, M. Roca-Martí, A. M. Wyatt, L. Resplandy, S. J. Clevenger, J. A. Drysdale, M. L. Estapa, S. Pike, and B. P. Umhau: High-resolution spatial and temporal measurements of particulate organic carbon flux using thorium-234 in the northeast Pacific Ocean during the EXport Processes in the Ocean from RemoTe Sensing field campaign, *Elementa-Sci Anthrop*, 8, 1, 10.1525/elementa.2020.030, 2020.

Cai, P. H., D. C. Zhao, L. Wang, B. Q. Huang, and M. H. Dai: Role of particle stock and phytoplankton community structure in regulating particulate organic carbon export in a large marginal sea, *J Geophys Res-Oceans*, 120, 2063-2095, 10.1002/2014jc010432, 2015.

Gan, J. P., Z. Q. Liu, and L. L. Liang: Numerical modeling of intrinsically and extrinsically forced seasonal circulation in the China Seas: A kinematic study, *J Geophys Res-Oceans*, 121, 4697-4715, 10.1002/2016jc011800, 2016.

Guo, M. X., P. Xiu, S. Y. Li, F. Chai, H. J. Xue, K. B. Zhou, and M. H. Dai: Seasonal variability and mechanisms regulating chlorophyll distribution in mesoscale eddies in the South China Sea, *J Geophys Res-Oceans*, 122, 5329-5347, 10.1002/2016jc012670, 2017.

Ma, W. T., P. Xiu, F. Chai, L. H. Ran, M. G. Wiesner, J. Y. Xi, Y. W. Yan, and E. Fredj: Impact of mesoscale eddies on the source funnel of sediment trap measurements in the South China Sea, *Prog Oceanogr*, 194, 10.1016/j.pocean.2021.102566, 2021.

McGillicuddy, D. J., L. A. Anderson, S. C. Doney, and M. E. Maltrud: Eddy-driven sources and sinks of nutrients in the upper ocean: Results from a 0.1 degrees resolution model of the North Atlantic, *Global Biogeochem Cy*, 17, 10.1029/2002gb001987, 2003.

Wang, L., B. Q. Huang, E. A. Laws, K. B. Zhou, X. Liu, Y. Y. Xie, and M. H. Dai: Anticyclonic Eddy Edge Effects on Phytoplankton Communities and Particle Export in the Northern South China Sea, *J Geophys Res-Oceans*, 123, 7632-7650, 10.1029/2017jc013623, 2018.

Wei, C. L., S. Y. Lin, D. D. Sheu, W. C. Chou, M. C. Yi, P. H. Santschi, and L. S. Wen: Particle-reactive radionuclides ([234]Th, [210]Pb, [210]Po) as tracers for the estimation of export production in the South China Sea, *Biogeosciences*, 8, 3793-3808, 10.5194/bg-8-3793-2011, 2011.

170    Zhao, D. D., Y. S. Xu, X. G. Zhang, and C. Huang: Global chlorophyll distribution induced by mesoscale eddies, *Remote Sens Environ*, 254, 10.1016/j.rse.2020.112245, 2021.

Zhou, K. B., M. H. Dai, K. Maiti, W. F. Chen, J. H. Chen, Q. Q. Hong, Y. F. Ma, P. Xiu, L. Wang, and Y. Y. Xie: Impact of physical and biogeochemical forcing on particle export in the South China Sea, *Prog Oceanogr*, 187, 102403, 10.1016/j.pocean.2020.102403, 2020.

175

**Anonymous Referee #2**

The article submitted by Ma et al investigates carbon export from the euphotic layer of the South China Sea, considering two layers (nutricline and euphotic layers). Carbon exports were calculated based on $^{234}$Th particulate fluxes and POC:$^{234}$Th ratio. The authors made a complicated discussion on the potential origin of nitrogen sources based on $^{15}$N-isotopic budget. The article
180    is rather difficult to follow as the description of the dataset is not clear, and the sections are not always in the appropriate order. For example, it is really strange to discuss the impacts on physical transport on $^{234}$Th fluxes, whereas there were several pages where the $^{234}$Th fluxes and derived product were extensively discussed.

[Response]: We appreciate the constructive comments from the reviewer. Our manuscript has been thoroughly revised
185    according to the reviewer's comments to optimize the discussion and the logic flow, and to enhance the readability. To do so, we have made a new table (Table R1) to better describe the dataset being used as suggested by the reviewer. In addition, the discussion of the physical transport on $^{234}$Th flux (section 4.1) will be moved to the methods part, before the $^{234}$Th derived fluxes are discussed.

190

Table R1: Sampling logs and site information along with the accessed parameters and their utilizations.

| Station | Arriving time | Latitude [ºN] | Longitude [ºE] | Bottom depth [m] | Parameters | | Data utilizations | |
| | | | | | Total $^{234}$Th | Trap | Partitioning POC flux estimate | Nutrient source diagnosis |
|---|---|---|---|---|---|---|---|---|
| SEATS | 2017-06-07 00:06 | 18 | 116 | 3907 | √ | √ | √ | √ |
| A1* | 2017-06-11 23:55 | 16 | 116 | 4205 | √ | | √ | |
| SS1 | 2017-06-12 20:08 | 14 | 116 | 4107 | √ | | √ | |
| H06 | 2017-06-20 02:28 | 14.1 | 116 | 4289 | √ | | √ | |
| H08 | 2017-06-20 07:51 | 13.9 | 116 | 4063 | √ | | √ | |
| H01 | 2017-06-20 23:41 | 14 | 116.1 | 4139 | √ | | √ | |
| H11 | 2017-06-21 05:18 | 14 | 115.9 | 4297 | √ | | √ | |
| B1 | 2017-06-22 11:43 | 14 | 113 | 2537 | √ | | √ | |
| C1 | 2017-06-23 04:40 | 12 | 113 | 4313 | √ | | √ | |
| A2 | 2017-06-24 03:05 | 12 | 116 | 4079 | √ | | √ | |
| B2 | 2017-06-24 21:42 | 14 | 117 | 3947 | √ | | √ | |

195 * Sampling station might be influenced by the typhoon event passing through the South China Sea. Station A1 was visited after typhoon Merbok, which was generated on June 9, 2017 at 13.1ºN, 119.8ºE in the southern South China Sea. Merbok landed on June 12 at 27.5ºN, 117.3ºE.

The dataset: there is a need of a table that presents clearly sampling, which station / when / what was measured (water column,

200  trap). It is indicated that the cruise took place from June, 5 to 27, 2017. But typhoon Merbok occurred the 10th. "before our field campaign". This needs to be clarified. In case of a typhoon had occurred during sampling, one could expect it had impacted the water column and chemical budget. In addition, how could it be possible to use the described $^{234}$Th model which is a steady-state model

[Response]: Thanks for the advice from the reviewer, and a new table of the sampling information will be included in the MS
205  and is shown above.

Note that we did not conduct samplings before or during the typhoon, thus it is impossible for us to build up a non-steady state model for $^{234}$Th flux estimation. However, we reasoned that a steady state model is in order in the condition under study as the Chl $a$ concentration was not significantly enhanced under the impact of the typhoon as shown by the remote sensing derived 8-day averaged surface Chl $a$ (Fig. R1). Additional justification of the steady state assumption has been added in line
210  293 during the revision: "We found that the $^{234}$Th fluxes remained rather low, mostly <800 dpm m$^{-2}$ d$^{-1}$ during our study, which were close to the threshold for the validity of steady-state assumption as shown in many prior studies (e.g., Savoye et al., 2006; Resplandy et al., 2012). The sea surface Chl $a$ also indicated that no bloom was observed during the survey in Jun. 2017 (Fig. S4), suggesting that the study area retained its biogeochemistry under the steady state condition."

[Figure]

**Figure R1:** Satellite-derived the 8-day averaged surface Chl *a* in the SCS basin during June 2017, showing that the sea surface Chl *a* concentration was little enhanced during our ship-based sampling period. Note that Station A1 was visited after typhoon Merbok, which was generated on June 9, 2017 at 13.1°N, 119.8°E in the southern South China Sea. Merbok landed on June 12 at 27.5°N, 117.3°E.

215

220    Four stations (H01, H06, H08, H11) were sampled around SS1 to check the spatial variability of $^{234}$Th. But it is indicated later that the mega station SS1 was revisited during August 2019 (after a second typhoon Mun, July, 1th) with trap deployment. A clarification must then to be made on what was sampled / when / where.

**[Response]:** We apologize that we were not clear enough in describing the multiple events that happened prior to and post our
225    sampling campaign. We have now included such information in Table R1 with clarifications throughout the revised MS. Note that our ship-based sampling occurred from June 5$^{th}$ to June 27$^{th}$, 2017 with samplings at station SS1 and its surrounding stations (H01, H06, H08 and H11) on June 12$^{th}$, 2017. We did deploy sediment traps at Station SS1 but unfortunately, the traps were not retrieved. We thus used trap results deployed two years later on July 13$^{th}$, 2019 accessed from Station SS1. It must be pointed out that the data accessed from sediment traps deployed at Station SS1 in 2019 was only utilized to evaluate the
230    contribution of subsurface nutrients by $\delta^{15}N_{PN}$. We add the text: "We visited two mega stations (SEATS and SS1) and 9 regular stations during the cruise. The *in situ* observation at Station SEATS was conducted before a typhoon (Merbok) which potentially affected the biogeochemistry of the region, and the remaining stations were visited after the typhoon (listed in Table 1). To examine the spatial variability of $^{234}$Th, we sampled four closely-clustered stations (H01, H06, H08, and H11) around Station SS1." in line 85 to clarify the sampling strategies in the revision.
235

High resolutions profiles: the authors made a great announcement about high resolution profiles. In fact, there are only two, t detailed profiles: SEAT and SS1. The other profiles have a less resolution, and, except for the lower total $^{234}$Th values at about 25 meters at station A1, the profiles of total $^{234}$Th are not so different. It would be interesting that the authors reduce the depth resolution of the SEATS and SS1 profiles to compare the estimated $^{234}$Th.
240

    **[Response]:** The reviewer is right that the 10-m vertical interval samplings were only conducted at stations SEATS and SS1. We will clarify this in our revision. Following suggestions, we add the text in line 299: "Given that our high vertical resolution of sampling mode was only applied to stations SEATS and SS1, we estimated $^{234}$Th fluxes at the Ez base by reducing the vertical resolution to a 25-m interval so as to be consistent with other stations, This exercise resulted in values of 490±60
245    and 655±71 dpm m$^{-2}$ d$^{-1}$ respectively for stations SEATS and SS1 compared to 522±43 and 631±48 dpm m$^{-2}$ d$^{-1}$ under the high-resolution sampling mode. The low-resolution sampling thus might induce an uncertainty of less than 6% for the $^{234}$Th flux. However, high-resolution sampling is essential in order to examine the partitioning of carbon export in the upper water column, especially for the oligotrophic ocean characteristic of low export fluxes.

    Based on the high-resolution total $^{234}$Th pattern at stations SEATS and SS1, we first determined $^{234}$Th deficit in the NDL,
250    showing substantial particle scavenging and POC export at the NDL base at both stations, and we subsequently found similar patterns at the rest of stations where estimated the partitioning in POC export fluxes."

The reduced sampling resolution might introduce some additional uncertainty to estimates of $^{234}$Th flux and $^{234}$Th-derived POC export flux, but would not change our main conclusion that the base of NDL is the hotspot for particle scavenging and POC export. We will include the above clarification and reasoning in our revision.

255

Export model: from equation (2), the authors need to produce the two equations relative to the export estimate for the NDL- and NRL-layers, respectively. Use directly the symbol for Fndl and Fnrl. There is no need to use layer i /i-1, that only complicate the model presentation. Also from Fig 2, it seems that calculations are done for each box, but from the text it is less clear that the fluxes from NRL-layer is calculated considering only the lower box or the whole water column above the euphotic

260 layer limit. Figure 2 needs also to be improved: if total $^{234}$Th activities are related to U activities, what means 'absorb particles, total TH already includes particulate phase. The figure needs to be corrected.

[Response]: We agree with the reviewer's comments for Figure 2. We actually calculated $^{234}$Th flux at the export horizons of the NDL base and the euphotic zone (Ez) bottom, with the integration carried out between the 0-NDL base and 0-Ez bottom

265 (not the lower box as mentioned by the reviewer). Here we use symbols $F_{NDL}$ and $F_{Ez}$ as suggested by the reviewer. To make the statement clearer, we will revise the main text to emphasize that the flux at the Ez bottom is integrated from the whole box from the surface to the Ez bottom.

The reviewer is also right that we only measured total $^{234}$Th activities during the cruises, and we will delete the "particles" in the figure and change "$A^{dissolved}$" into "$A^{total}$" as suggested by the reviewer (see the details in Fig. R2).

[Figure]

270

**Figure R2:** Schematic of the $^{234}$Th model under the two-layer nutrient structure. All terms are defined in Equations (2)-(4) and (7)-(9).

275 The conversion of $^{234}$Th particulate fluxes in POC fluxes: the conversion is done using the POC/$^{234}$Th. The recommendation is to use the large particle ratio. In this work, the authors use the ratio obtained from bottle waters, that correspond to fine particles. The authors need to better argument the choice. The comparison with the trap ratio seems to be biased as trap was

done in summer, no during the same sampling cruise. The authors need to be clearer on this aspect. If confirmed, it means that some paragraphs are not justified.

280

[Response]: Bottle filtration and trap deployment for POC/$^{234}$Th were done at Station SEATS during the same cruise (See Table R1). Bottle-derived POC/$^{234}$Th ratios at the depth of 50 m and 100 m were respectively 4.4±0.6 and 3.8±0.3 µmol C dpm$^{-1}$ compared to 4.4±0.6 and 3.2±0.4 µmol C dpm$^{-1}$ from trap samples. We thus confirmed that bottle-derived POC/$^{234}$Th was comparable with those derived from trap samples during this cruise. This is consistent with what Zhou et al. (2020) found

285    showing that POC export fluxes based on bottle POC/$^{234}$Th were comparable with trap POC fluxes measured before. More importantly, it was impossible to deploy sediment traps at all stations due to practical reasons. For consistency with prior studies in the region (e.g., Cai et al., 2008; Zhou et al., 2013; Cai et al., 2015; Zhou et al., 2020), we primarily used bottle-derived POC/$^{234}$Th in estimating POC export fluxes as we routinely did in our prior work. To clarify the POC/$^{234}$Th ratios we employed in the POC export flux estimates, we also add the text in line 382 during the revision: "We thus confirmed that

290    bottle-derived POC/$^{234}$Th was comparable with those derived from sinking particles accessed from traps or *in situ* pumps. This is consistent with prior studies showing that POC export fluxes based on bottle POC/$^{234}$Th were comparable with trap POC fluxes (e.g., Zhou et al., 2020a)"

Th/POC flux estimates: most of the article is based on fluxes, but the authors treated data as it was rather instantaneous fluxes,

295    which is clearly not the case. Considering the half-life of $^{234}$Th, a deficit of $^{234}$Th in the water column represent a flux story of several weeks. The only way to have more "instantaneous" fluxes is to repeat profiles at the same station which was not done here. Therefore, it is the main problem with the article. The authors discussed a lot fluxes and potential nutrient sources, but the errors on the fluxes estimate do not support the discussion. There is an over-interpretation of the dataset and the derived fluxes to support the hypothesis of the authors.

300

[Response]: We completely agree with the reviewer that $^{234}$Th-derived POC export flux is not instantaneous but with a timescale of weeks to months. In order to match the time scale between nutrient and POC fluxes, we also correlated $^{234}$Th-derived POC flux with the model-derived monthly average of nutrients during summer (Fig. R3, Du et al., 2021). The correlations are indeed statistically significant (P<0.05). The additional text would be appended at the end of Section 4.3.1: "It

305    is also noteworthy that the timescale of ship-based nutrients data is instantaneous, which may differ from the timescale of $^{234}$Th method of weeks to months. Consequently, the correlations between *in situ* nutrients and $^{234}$Th-derived POC fluxes may be misinterpreted by the difference in timescales. To further investigate the correlations between nutrients and $^{234}$Th-derived POC fluxes, $^{234}$Th-derived POC fluxes were also related to the model-derived monthly average of nutrients (i.e., nutrient concentration and the depth of nutricline, Du et al., 2021) during summer (Fig. S5). The correlations between the two

310    parameters showed to be statistically significant (P<0.05), again implying the importance of nutrient modulation on export fluxes." This suggests that under the oligotrophic condition of the present study, the euphotic layer is characterized by low

biological productivity, and the system in this study is pretty much in a steady state. The overall low [234]Th flux as we explained in our above responses to the reviewer, also supports this notion. In addition, we examined the $\delta^{15}N_{PN}$ value measured in several previous studies in the region (e.g., Kao et al., 2012; Yang et al., 2017; Yang et al., 2022). Taken together, we contend

315   that the conclusion of the subsurface nutrient supported largely is a well plausible interpretation of the dataset. Having said that, we will fully consider the comments from the reviewer and revise our MS accordingly.

[Figure]

**Figure R3:** Relationship between POC export fluxes at the NDL base (black dots) and Ez base (grey dots) vs. the model-derived depth of the top of the nutricline (top) and DIN concentration in the subsurface water at 100 m (bottom).

320

Others comments: most figures need to be improved and some data combined differently. What is the interest of figure 3?

**[Response]:** Thanks for the comments, we have revised the figures based on the suggestion above from the reviewer. As our
325   Fig. 4 has shown the vertical profiles of T and S, here we deleted Fig. 3 to simplify the discussion.

**References**

Cai, P. H., M. H. Dai, W. F. Chen, T. T. Tang, and K. B. Zhou: On the importance of the decay of [234]Th in determining size-fractionated C/[234]Th ratio on marine particles, *Geophys Res Lett*, 33, 23, 10.1029/2006gl027792, 2006.
330   Cai, P. H., D. C. Zhao, L. Wang, B. Q. Huang, and M. H. Dai: Role of particle stock and phytoplankton community structure in regulating particulate organic carbon export in a large marginal sea, *J Geophys Res-Oceans*, 120, 2063-2095, 10.1002/2014jc010432, 2015.

Cai, P. H., W. F. Chen, M. H. Dai, Z. W. Wan, D. X. Wang, Q. Li, T. T. Tang, and D. W. Lv: A high-resolution study of particle export in the southern South China Sea based on [234]Th : [238]U disequilibrium, *J Geophys Res-Oceans*, 113, C04019,
335   10.1029/2007jc004268, 2008.

Du, C. J., R. Y. He, Z. Y. Liu, T. Huang, L. F. Wang, Z. W. Yuan, Y. P. Xu, Z. Wang, and M. H. Dai: Climatology of nutrient distributions in the South China Sea based on a large data set derived from a new algorithm, *Prog Oceanogr*, 195, 102586, 10.1016/j.pocean.2021.102586, 2021.

340 Kao, S. J., Terence Yang, J. Y., Liu, K. K., Dai, M., Chou, W. C., Lin, H. L., Ren, H.: Isotope constraints on particulate nitrogen source and dynamics in the upper water column of the oligotrophic South China Sea. *Global Biogeochem Cycles*, 26: GB2033, 10.1029/2011GB004091, 2012.

Resplandy, L., Martin, A. P., Le Moigne, F., Martin, P., Aquilina, A., Mémery, L., Lévy, M. and Sanders, R.: How does dynamical spatial variability impact [234]Th-derived estimates of organic export? Deep Sea Research Part I, 68: 24-45, doi: 10.1016/j.dsr.2012.05.015, 2012.

345 Savoye, N., C. Benitez-Nelson, A. B. Burd, J. K. Cochran, M. Charette, K. O. Buesseler, G. A. Jackson, M. Roy-Barman, S. Schmidt, and M. Elskens: [234]Th sorption and export models in the water column: A review, *Mar Chem*, 100, 234-249, 10.1016/j.marchem.2005.10.014, 2006.

Yang, J. Y. T., Kao, S. J., Dai, M., Yan, X., Lin, H. L.: Examining N cycling in the northern South China Sea from N isotopic signals in nitrate and particulate phases. *J Geophys Res-Biogeoscience*, 122: 2118-2136, 10.1002/2016JG003618, 2017.

350 Yang, J. Y. T., Tang, J. M., Kang, S., Dai, M., Kao, S. J., Yan, X., Xu, M. N., Du, C.: Comparison of nitrate isotopes between the South China Sea and western North Pacific Ocean: Insights into biogeochemical signals and water exchange. *J Geophys Res-Oceans*, 127: e2021JC018304, 10.1029/2021JC018304, 2022.

Zhou, K. B., M. H. Dai, S. J. Kao, L. Wang, P. Xiu, F. Chai, J. W. Tian, and Y. Liu: Apparent enhancement of [234]Th-based particle export associated with anticyclonic eddies, *Earth Planet Sc Lett*, 381, 198-209, 10.1016/j.epsl.2013.07.039, 2013.

355 Zhou, K. B., M. H. Dai, K. Maiti, W. F. Chen, J. H. Chen, Q. Q. Hong, Y. F. Ma, P. Xiu, L. Wang, and Y. Y. Xie: Impact of physical and biogeochemical forcing on particle export in the South China Sea, *Prog Oceanogr*, 187, 102403, 10.1016/j.pocean.2020.102403, 2020.

---

## Author Response (AR2)

*Re: "Partitioning of carbon export in the euphotic zone of the oligotrophic South China Sea" by Ma et al.*

25th  April  2023

Dear Editor,

Thank you for your time in handling our paper. We are pleased to submit our further revised MS entitled "Partitioning of carbon export in the euphotic zone of the oligotrophic South China Sea" by Yifan Ma et al.

Throughout the revisions, we carefully considered the comments and suggestions from the reviewers. Specifically, we improved the abstract to better highlight our findings and changed the title from "upper water column" to "euphotic zone" to better reflect the study foci. Additionally, we thoroughly checked the manuscript for typos, as per the reviewers' comments.

Finally, we would like to take this opportunity to thank you and the reviewers for the comments and suggestions, which have significantly improved the quality of our paper. We sincerely hope that our revision meets the standards of *Biogeosciences*.

Sincerely,

Minhan Dai
Corresponding author
State Key Lab of Marine Environmental Science
Xiamen University
Xiamen 361005, China

**Anonymous Referee #1**

The revised manuscript is greatly improved. The result and interpretation are convincing. There are only some technical points that need to be corrected.

**[Response]:** We appreciate the Reviewer's positive feedback on our revisions.

In Eq. 2, the del sign before Ath total is a typo.

**[Response]:** Fixed. We have updated Eq. (2) as per the suggested modification as shown below:

$$\text{``}\frac{\partial A_{Th}^{total}}{\partial t} = \lambda \left( A_U - A_{Th}^{total} \right) - F_{Th} + V \qquad (2)\text{''}$$

L495, 48% is almost the half. It seems hard to suggest a major role of subsurface nutrients. Here needs some clarification.

**[Response]:** In the revised manuscript, we have further clarified this issue, which now reads: "…is estimated to be 48%, suggesting that 52% of PN flux at this depth is supported by subsurface nitrate. The derived $f_{NO_3^-}$ based on mass balance is slightly lower than that obtained from the isotopic balance at the NDL base (59-67%). This might be due to an overestimation of the nitrogen fixation rate and the flux of atmospheric nitrogen deposition in the mass balance model. For example, the nitrogen fixation rate used is observed in the northeastern SCS where the Kuroshio intrudes frequently (Kao et al., 2012). Higher rates of nitrogen fixation were detected in the Kuroshio-influenced waters compared to those in the northern basin (e.g., at SEATS station; Lu et al., 2019). Similarly, the observed flux of atmospheric nitrogen deposition at Dongsha Island, which is close to mainland China, is likely higher than that at the station SEATS. Despite uncertainties, the two independent estimates both suggest a substantial role of subsurface nitrate in supporting particle export out of the NDL base.".

**Anonymous Referee #2**

The authors greatly improved the manuscript. However it remains details that need to be improved.

**[Response]:** We express our appreciation for the positive feedback from the reviewer. We have made further revisions following the Reviewer's comments as elaborated in our responses provided below.

- update the state of the art line 48 to 53,

**[Response]:** In the revised manuscript we have updated some recently published papers, as detailed below:

"...such as [234]Th. Cai et al. (2008) also observed variable particle scavenging rates in the upper euphotic zone (above 50 m) but consistently lower rates in the lower euphotic zone (between 50 and 100 m) in the oligotrophic SCS. With increasing high-resolution samplings, such partitionings of [234]Th-based particle scavenging were frequently observable in oligotrophic ecosystems (Buesseler et al., 2009; Umhau et al., 2019; Zhou et al., 2020; Stukel et al., 2022)."

- improve the abstract

**[Response]:** Following the Reviewer's comment, we have improved the abstract to clearly show our findings and the implication of the study.

- improve the title that does not reflect the content of the article, "upper water column" is not enough precise regarding the objectives of the work

**[Response]:** As our study primarily examines the partitioning of carbon export within the euphotic zone, we will revise the title to "Partitioning of carbon export in the euphotic zone of the oligotrophic South China Sea".

- figure 2: again to improve, it is not total [234]Th that is exported but particulate Th, then AU --> AThdissolved --> ATHparticulate

**[Response]:** We have incorporated the terms " $A_{Th}^{\text{dissolved}}$ " and " $A_{Th}^{\text{particulate}}$ " to represent the total [234]Th in the Figure 2 in our revised manuscript. Arrows are placed directly below the $A_{Th}^{\text{particulate}}$ to

denote that the particles, including the $^{234}$Th in particulate phase are scavenging.

[Figure]

**Figure 2**: Schematic of the $^{234}$Th model under the two-layer nutrient structure. The terms are defined in Eq. (2)-(4) and Eq. (7)-(9).

**References**

Buesseler, K. O., Pike, S., Maiti, K., Lamborg, C. H., Siegel, D. A., and Trull, T. W.: Thorium-234 as a tracer of spatial, temporal and vertical variability in particle flux in the North Pacific, *Deep-Sea Res Part I*, 56, 1143-1167, 10.1016/j.dsr.2009.04.001, 2009.

Cai, P. H., Chen, W. F., Dai, M. H., Wan, Z. W., Wang, D. X., Li, Q., Tang, T. T., and Lv, D. W.: A high-resolution study of particle export in the southern South China Sea based on $^{234}$Th : $^{238}$U disequilibrium, *J Geophys Res-Oceans*, 113, C04019, 10.1029/2007jc004268, 2008.

Kao, S. J., Yang, J. Y. T., Liu, K. K., Dai, M. H., Chou, W. C., Lin, H. L., and Ren, H. J.: Isotope constraints on particulate nitrogen source and dynamics in the upper water column of the oligotrophic South China Sea, *Global Biogeochem Cycles*, 26, (2), 10.1029/2011gb004091, 2012.

Lu, Y. Y., Wen, Z. Z., Shi, D. L., Lin, W. F., Bonnet, S., Dai, M. H., and Kao, S. J.: Biogeography of $N_2$ fixation influenced by the western boundary current intrusion in the South China Sea, *J Geophys Res-Oceans*, 124, 6983-6996, 10.1029/2018JC014781, 2019.

Stukel, M. R., Kelly, T. B., Landry, M. R., Selph, K. E., and Swalethorp, R.: Sinking carbon, nitrogen, and pigment flux within and beneath the euphotic zone in the oligotrophic, open-ocean Gulf of Mexico, *J Plankton Res*, 44, 711-727, 10.1093/plankt/fbab001, 2022.

Umhau, B. P., Benitez-Nelson, C. R., Close, H. G., Hannides, C. C. S., Motta, L., Popp, B. N., Blum, J. D., and Drazen, J. C.: Seasonal and spatial changes in carbon and nitrogen fluxes estimated using $^{234}$Th:$^{238}$U disequilibria in the North Pacific tropical and subtropical gyre, *Mar Chem*, 217, 103705, 10.1016/j.marchem.2019.103705, 2019.

Zhou, K. B., Dai, M. H., Maiti, K., Chen, W. F., Chen, J. H., Hong, Q. Q., Ma, Y. F., Xiu, P., Wang, L., and Xie, Y. Y.: Impact of physical and biogeochemical forcing on particle export in the South China Sea, *Prog Oceanogr*, 187, 102403, 10.1016/j.pocean.2020.102403, 2020.